# 💥Dataset Distillation via Curriculum Data Synthesis in Large Data Era

**Zeyuan Yin**                                                    *zeyuan.yin@mbzuai.ac.ae*
*VILA Lab*
*Mohamed bin Zayed University of Artificial Intelligence*

**Zhiqiang Shen**[*]                                              *zhiqiang.shen@mbzuai.ac.ae*
*VILA Lab*
*Mohamed bin Zayed University of Artificial Intelligence*

**Reviewed on OpenReview:** *https://openreview.net/forum?id=PlaZD2nGCl*

## Abstract

Dataset distillation or condensation aims to generate a smaller but representative subset from a large dataset, which allows a model to be trained more efficiently, meanwhile evaluating on the original testing data distribution to achieve decent performance. Previous decoupled methods like SRe$^2$L simply use a unified gradient update scheme for synthesizing data from Gaussian noise, while, we notice that the initial several update iterations will determine the final outline of synthesis, thus an improper gradient update strategy may dramatically affect the final generation quality. To address this, we introduce a simple yet effective *global-to-local* gradient refinement approach enabled by curriculum data augmentation (`CDA`) during data synthesis. The proposed framework achieves the current published highest accuracy on both large-scale ImageNet-1K and 21K with 63.2% under IPC (Images Per Class) 50 and 36.1% under IPC 20, using a regular input resolution of 224×224 with faster convergence speed and less synthetic time. The proposed model outperforms the current state-of-the-art methods like SRe$^2$L, TESLA, and MTT by more than 4% Top-1 accuracy on ImageNet-1K/21K and for the first time, reduces the gap to its full-data training counterparts to less than absolute 15%. Moreover, this work represents the inaugural success in dataset distillation on the larger-scale ImageNet-21K dataset under the standard 224×224 resolution. Our code and distilled ImageNet-21K dataset of 20 IPC, 2K recovery budget are available at `https://github.com/VILA-Lab/SRe2L/tree/main/CDA`.

## 1 Introduction

Dataset distillation or condensation (Wang et al., 2018) has attracted considerable attention across various fields of computer vision (Cazenavette et al., 2022b; Cui et al., 2023; Yin et al., 2023) and natural language processing (Sucholutsky & Schonlau, 2021; Maekawa et al., 2023). This task aims to optimize the process of condensing a massive dataset into a smaller, yet representative subset, preserving the essential features and characteristics that would allow a model to learn from scratch as effectively from the distilled dataset as it would from the original large

Figure 1: ImageNet-1K comparison with SRe$^2$L.

---

[*]Corresponding author.

dataset. As the scale of data and models continue to grow, this *dataset distillation* concept becomes even more critical in the large data era, where datasets are often voluminous that they pose storage, computational, and processing challenges. Generally, dataset distillation can level the playing field, allowing researchers with limited computation and storage resources to participate in state-of-the-art foundational model training and application development, such as affordable ChatGPT (Brown et al., 2020; OpenAI, 2023) and Stable Diffusion (Rombach et al., 2022), in the current large data and large model regime. Moreover, by working with distilled datasets, which are synthesized to retain the most representative information from Gaussian noise initialization through gradient optimization at a high-level abstraction instead of closely resembling the original dataset, there is potential to alleviate data privacy concerns, as raw, personally identifiable data points might be excluded from the distilled version.

Recently, there has been a significant trend in adopting large models and large data across various research and application areas. Yet, many prior dataset distillation methods (Wang et al., 2018; Zhao et al., 2020; Zhou et al., 2022; Cazenavette et al., 2022a; Kim et al., 2022a; Cui et al., 2023) predominantly target datasets like CIFAR, Tiny-ImageNet and downsampled ImageNet-1K, finding it challenging to scale their frameworks for larger datasets, such as full ImageNet-1K (Deng et al., 2009). This suggests that these approaches have not fully evolved in line with contemporary advancements and dominant methodologies.

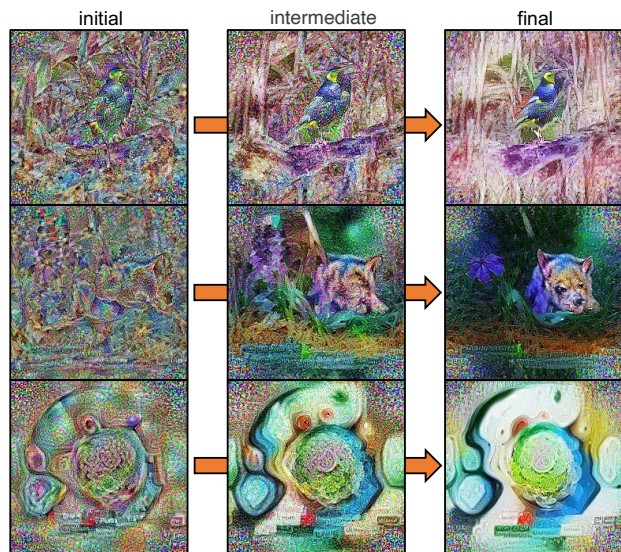

Figure 2: Motivation of our work. The left column is the synthesized images after a few gradient update iterations from Gaussian noise. Middle and right columns are intermediate and final synthesized images.

In this study, we extend our focus even beyond the ImageNet-1K dataset, venturing into the uncharted territories of the full ImageNet-21K (Deng et al., 2009; Ridnik et al., 2021) at a conventional resolution of 224×224. This marks a pioneering effort in handling such a vast dataset for dataset distillation task. Our approach harnesses a straightforward yet effective global-to-local learning framework. We meticulously address each aspect and craft a robust strategy to effectively train on the complete ImageNet-21K, ensuring comprehensive knowledge is captured. Specifically, following a prior study (Yin et al., 2023), our approach initially trains a model to encapsulate knowledge from the original datasets within its dense parameters. However, we introduce a refined training recipe that surpasses the results of Ridnik et al. (2021) on ImageNet-21K. During the data recovery/synthesis phase, we employ a strategic learning scheme where partial image crops are sequentially updated based on the difficulty of regions: transitioning either from simple to difficult, or vice versa. This progression is modulated by adjusting the lower and upper bounds of the *RandomReiszedCrop* data augmentation throughout varying training iterations. Remarkably, we observe that this straightforward learning approach substantially improves the quality of synthesized data. In this paper, we delve into three learning paradigms for data synthesis linked to the curriculum learning framework. The first is the standard curriculum learning, followed by its alternative approach, reverse curriculum learning. Lastly, we also consider the basic and previously employed method of constant learning.

**Motivation and Intuition.** We aim to maximize the global informativeness of the synthetic data. Both SRe$^2$L (Yin et al., 2023) and our proposed approach utilize local mini-batch data's mean and variance statistics to match the global statistics of the entire original dataset, synthesizing data by applying gradient updates directly to the image. The impact of such a strategy is that the initial few iterations set the stage for the global structure of the ultimately generated image, as shown in Figure 2. Building upon the insights derived from the analysis, we can leverage the *global-to-local* gradient refinement scheme for more expressive synthesized data, in contrast, SRe$^2$L does not capitalize on this characteristic. Specifically, our proposed approach exploits this by initially employing large crops to capture a more accurate and complete outline of

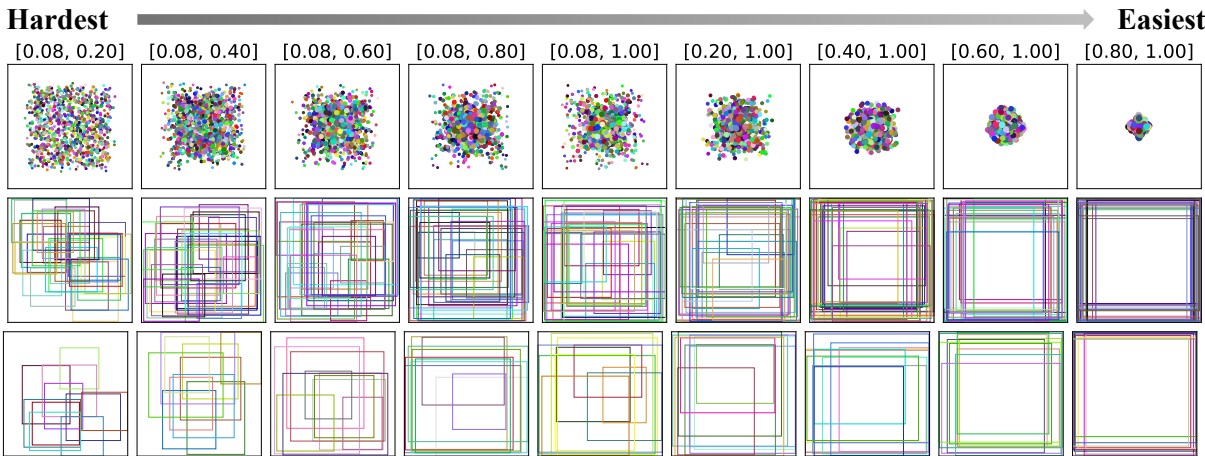

Figure 3: Illustration of crop distribution from different lower and upper bounds in *RandomResizedCrop*. The first row is the central points of bounding boxes from different sampling scale hyperparameters. The second and last rows correspond to 30 and 10 boxes of the crop distributions. In each row, from left to right, the difficulty of crop distribution is decreasing.

objects, for building a better foundation. As the process progresses, it incrementally reduces the crop size to enhance the finer, local details of the object, significantly elevating the quality of the synthesized data.

**Global-to-local via Curriculum Sampling.** *RandomResizedCrop* randomly crops the image to a certain area and then resizes it back to the pre-defined size, ensuring that the model is exposed to different regions and scales of the original image during training. As illustrated in Figure 3, the difficulty level of the cropped region can be controlled by specifying the lower and upper bounds for the area ratio of the crop. This can be used to ensure that certain portions of the image (small details or larger context) are present in the cropped region. If we aim to make the learning process more challenging, reduce the minimum crop ratio. This way, the model will often see only small portions of the image and will have to learn from those limited contexts. If we want the model to see a larger context more frequently, increase the minimum crop ratio. In this paper, we perform a comprehensive study on how the gradual difficulty changes by sampling strategy influence the optimization of data generation and the quality of synthetic data for dataset distillation. Our proposed curriculum data augmentation (CDA) is a heuristic and intuitive approach to simulate a *global-to-local* learning procedure. Moreover, it is highly effective on large-scale datasets like ImageNet-1K and 21K, achieving state-of-the-art performance on dataset distillation.

**The Significance of Large-scale Dataset Condensation.** Large models trained on large-scale datasets consistently outperform smaller models and those trained on limited data (Dosovitskiy et al., 2020; Dehghani et al., 2023; OpenAI, 2023). Their ability to capture intricate patterns and understand nuanced contextual information makes them exceptionally effective across a wide range of tasks and domains. These models play a crucial role in solving complex industrial challenges and accelerating the development of AI-driven products and services, thereby contributing to economic growth and innovation. Therefore, adapting dataset condensation or distillation methods for large-scale data scenarios is vital to unlocking their full potential in both academic and industrial applications.

We conduct extensive experiments on the CIFAR, Tiny-ImageNet, ImageNet-1K, and ImageNet-21K datasets. Employing a resolution of 224×224 and IPC 50 on ImageNet-1K, the proposed approach attains an impressive accuracy of 63.2%, surpassing all prior state-of-the-art methods by substantial margins. As illustrated in Figure 1, our proposed CDA outperforms SRe$^2$L by 4∼6% across different architectures under 50 IPC, on both 1K and 4K recovery budgets. When tested on ImageNet-21K with IPC 20, our method achieves a top-1 accuracy of 35.3%, which is closely competitive, exhibiting only a minimal gap compared to the model pre-trained with full data, at 44.5%, while using 50× fewer training samples.

Our contributions of this work:

- We propose a new curriculum data augmentation (CDA) framework enabled by *global-to-local* gradient update in data synthesis for large-scale dataset distillation.

- We are the first to distill the ImageNet-21K dataset, which reduces the gap to its full-data training counterparts to less than an absolute 15% accuracy.

- We conduct extensive experiments on CIFAR-100, Tiny-ImageNet, ImageNet-1K and ImageNet-21K datasets to demonstrate the effectiveness of the proposed approach.

## 2 Related Work

Dataset condensation or distillation strives to form a compact, synthetic dataset, retaining crucial information from the original large-scale dataset. This approach facilitates easier handling, reduces training time, and aims for performance comparable to using the full dataset. Prior solutions typically fall under four categories: *Meta-Model Matching* optimizes for model transferability on distilled data, with an outer-loop for synthetic data updates, and an inner-loop for network training, such as DD (Wang et al., 2020), KIP (Nguyen et al., 2021), RFAD (Loo et al., 2022), FRePo (Zhou et al., 2022), LinBa (Deng & Russakovsky, 2022), and MDC (He et al., 2024); *Gradient Matching* performs a one-step distance matching between models, such as DC (Zhao et al., 2020), DSA (Zhao & Bilen, 2021), DCC (Lee et al., 2022), IDC (Kim et al., 2022b), and MP (Zhou et al., 2024a); *Distribution Matching* directly matches the distribution of original and synthetic data with a single-level optimization, such as DM (Zhao & Bilen, 2023), CAFE (Wang et al., 2022), HaBa (Liu et al., 2022a), KFS (Lee et al., 2022), DataDAM (Sajedi et al., 2023), FreD Shin et al. (2024), and GUARD (Xue et al., 2024); *Trajectory Matching* matches the weight trajectories of models trained on original and synthetic data in multiple steps, methods include MTT (Cazenavette et al., 2022b), TESLA (Cui et al., 2023), APM (Chen et al., 2023), and DATM (Guo et al., 2024).

Moreover, there are some recent methods out of these categories that have further improved the existing dataset distillation. SeqMatch (Du et al., 2023) reorganizes the synthesized dataset during the distillation and evaluation phases to extract both low-level and high-level features from the real dataset, which can be integrated into existing dataset distillation methods. Deep Generative Prior (Cazenavette et al., 2023) utilizes the learned prior from the pre-trained deep generative models to synthesize the distilled images. RDED (Sun et al., 2024) proposes a non-optimization method to concatenate multiple cropped realistic patches from the original data to compose the distilled dataset. D3M (Abbasi et al., 2024) condenses an entire category of images into a single textual prompt of latent diffusion models. SC-DD (Zhou et al., 2024b) proposes a self-supervised paradigm by applying the self-supervised pre-trained backbones for dataset distillation. EDC (Shao et al., 2024b) explores a comprehensive design space that includes multiple specific, effective strategies like soft category-aware matching and learning rate schedule to establishe a benchmark for both small and large-scale dataset distillation. Ameliorate Bias (Cui et al., 2024) studies the impact of bias within the original dataset on the performance of dataset condensation. It introduces a simple yet effective approach based on a sample reweighting scheme that utilizes kernel density estimation.

SRe$^2$L (Yin et al., 2023) is the first and mainstream framework to distill large-scale datasets, such as ImageNet-1K, and achieve significant performance. Thus, we consider it as our closest baseline. More specifically, SRe$^2$L proposes a decoupling framework to avoid the bilevel optimization of model and synthesis during distillation, which consists of three stages of squeezing, recovering, and relabeling. In the first squeezing stage, a model is trained on the original dataset and serves as a frozen pre-train model in the following two stages. During the recovering stage, the distilled images are synthesized with the knowledge recovered from the pre-train model. At the last relabeling stage, the soft labels corresponding to synthetic images are generated and saved by leveraging the pre-train model. Recently, distilling on large-scale datasets has received significant attention in the community, and many works have been proposed, including (Sun et al., 2023; Liu et al., 2023; Chen et al., 2023; Shao et al., 2024a; Zhou et al., 2024a; Wu et al., 2024; Abbasi et al., 2024; Zhou et al., 2024b; Shao et al., 2024b; Xue et al., 2024; Qin et al., 2024; Gu et al., 2023; Ma et al., 2024; Shang et al., 2024). Theses recent methods represent a comprehensive study and literature on framework design space (Shao et al., 2024b) and adversarial robustness benchmarks (Wu et al., 2024) in dataset distillation. They are substantially different from our input-optimization-based approach. Additionally, GUARD (Xue et al., 2024) incorporates curvature regularization to embed adversarial robustness, focusing on a different objective than our CDA.

Compared to earlier traditional dataset distillation baselines, CDA is fundamentally different in its approach: (1) CDA exhibits better scalability. The previous works like DM (Zhao & Bilen, 2023), DSA (Zhao & Bilen, 2021), and FRePo (Zhou et al., 2022), work well on small-scale dataset distillation, but they are limited by the huge computational cost and cannot be scaled to large datasets and models. (2) Different generation paradigms. MTT (Cazenavette et al., 2022a) matches the model trajectories (weights) of training on distilled and raw datasets; RDED (Sun et al., 2023) selects and combines the raw image patches with diversity; D3M (Abbasi et al., 2024) leverages text-to-image diffusion models to generate distilled images. Thus, MTT proposes matching trajectories, RDED proposes non-optimizing, and D3M proposes diffusion-model-based generation paradigms which do not align with to our knowledge-distillation-based generation approach. (3) Unique evaluation recipes. For instance, RDED utilizes a unique smoothed LR schedule for the learning rate reduction throughout the evaluation, which improves evaluation performance effectively[1].

## 3 Approach

### 3.1 Preliminary: Dataset Distillation

The goal of dataset distillation is to derive a concise synthetic dataset that maintains a significant proportion of the information contained in the original, much larger dataset. Suppose there is a large labeled dataset $\mathcal{D}_o = \left\{ (\boldsymbol{x}_1, \boldsymbol{y}_1), \ldots, (\boldsymbol{x}_{|\mathcal{D}_o|}, \boldsymbol{y}_{|\mathcal{D}_o|}) \right\}$, our target is to formulate a compact distilled dataset, represented as $\mathcal{D}_d = \left\{ (\boldsymbol{x}'_1, \boldsymbol{y}'_1), \ldots, (\boldsymbol{x}'_{|\mathcal{D}_d|}, \boldsymbol{y}'_{|\mathcal{D}_d|}) \right\}$, where $\boldsymbol{y}'$ is the soft label corresponding to synthetic data $\boldsymbol{x}'$, and $|\mathcal{D}_d| \ll |\mathcal{D}_o|$, preserving the essential information from the original dataset $\mathcal{D}_o$. The learning objective based on this distilled synthetic dataset is:

$$\boldsymbol{\theta}_{\mathcal{D}_d} = \arg\min_{\boldsymbol{\theta}} \mathcal{L}_{\mathcal{D}_d}(\boldsymbol{\theta}) \tag{1}$$

$$\mathcal{L}_{\mathcal{D}_d}(\boldsymbol{\theta}) = \mathbb{E}_{(\boldsymbol{x}', \boldsymbol{y}') \in \mathcal{D}_d} \Big[ \ell(\phi_{\boldsymbol{\theta}_{\mathcal{D}_d}}(\boldsymbol{x}'), \boldsymbol{y}') \Big] \tag{2}$$

where $\ell$ is the regular loss function such as the soft cross-entropy, and $\phi_{\boldsymbol{\theta}_{\mathcal{D}_d}}$ is model. The primary objective of the dataset distillation task is to generate synthetic data aimed at attaining a specific or minimal performance disparity on the original validation data when the same models are trained on the synthetic data and the original dataset, respectively. Thus, we aim to optimize the synthetic data $\mathcal{D}_d$ by:

$$\arg\min_{\mathcal{D}_d, |\mathcal{D}_d|} \left( \sup \left\{ \left| \ell\left(\phi_{\boldsymbol{\theta}_{\mathcal{D}_o}}(\boldsymbol{x}_{val}), \boldsymbol{y}_{val}\right) - \ell\left(\phi_{\boldsymbol{\theta}_{\mathcal{D}_d}}(\boldsymbol{x}_{val}), \boldsymbol{y}_{val}\right) \right| \right\}_{(\boldsymbol{x}_{val}, \boldsymbol{y}_{val}) \sim \mathcal{D}_o} \right) \tag{3}$$

where $(\boldsymbol{x}_{val}, \boldsymbol{y}_{val})$ are sample and label pairs in the validation set of the real dataset $\mathcal{D}_o$. Then, we learn $<\text{data}, \text{label}> \in \mathcal{D}_d$ with the corresponding number of distilled data in each class.

### 3.2 Dataset Distillation on Large-scale Datasets

Currently, the prevailing majority of research studies within dataset distillation mainly employ datasets of a scale up to ImageNet-1K (Cazenavette et al., 2022b; Cui et al., 2023; Yin et al., 2023) as their benchmarking standards. In this work, we are the pioneer in showing how to construct a strong baseline on ImageNet-21K (the approach is equivalently applicable to ImageNet-1K) by incorporating insights presented in recent studies, complemented by conventional optimization techniques. Our proposed baseline is demonstrated to achieve state-of-the-art performance over prior counterparts. We believe this provides substantial significance towards understanding the true impact of proposed methodologies on dataset distillation task and towards assessing the true gap with full original data training. We further propose a curriculum training paradigm to achieve a more informative representation of synthetic data. Following prior work in dataset distillation (Yin et al., 2023), we focus on the decoupled training framework, *Squeeze-Recover-Relabel*, to save computation and memory consumption on large-scale ImageNet-21K, the procedures are listed below:

**Squeeze: Building A Strong Pre-trained Model on ImagNet-21K**. To obtain a squeezing model, we use a relatively large label smooth of 0.2 together with Cutout (DeVries & Taylor, 2017) and RandAugment (Cubuk et al., 2020), as shown in Appendix B.4. This recipe helps achieve ∼2% improvement over the default training (Ridnik et al., 2021) on ImageNet-21K, as provided in Table 21.

---

[1]Therefore, to ensure a fair and straightforward comparison, these baseline results in our experimental section have been taken directly from the best evaluation performance reported by their original papers.

**Recover: Curriculum Training for Better Representation of Synthetic Data**. A well-crafted curriculum data augmentation is employed during the synthesis stage to realize the global-to-local learning scheme and enhance the representational capability of the synthetic data. This step is crucial, serving to enrich the generated images by embedding more knowledge accumulated from the original dataset, thereby making them more informative. Detailed procedures will be further described in the following Section 3.3.

**Relabel: Post-training on Larger Models with Stronger Training Recipes**. Prior studies, such as TESLA (Cui et al., 2023), have encountered difficulties, particularly, a decline in accuracy when utilizing models of larger scale. The reason may be that the trajectory-based matching approaches, e.g., MTT and TESLA, generate images by excessively optimizing to align the dense training trajectories of model weights at each epoch between real and distilled datasets on specific backbone models. As a result, the distilled dataset becomes overly dependent on these models, potentially leading to overfitting and reduced effectiveness when training other models, particularly larger ones. This further suggests that the synthetic data used is potentially inadequate for training larger models. Conversely, the data we relabel show improvement with the use of larger models combined with enhanced post-training methodologies, displaying promise when applied to larger datasets in distillation processes.

We have also observed that maintaining a smaller batch size is crucial for post-training on synthetic data to achieve commendable accuracy. This is attributed to the *Generalization Gap* (Keskar et al., 2016; Hoffer et al., 2017), which suggests that when there is a deficiency in the total training samples, the model's capacity to generalize to new, unseen data is not robust. In the context of synthetic data, the generalization gap can be exacerbated due to the inherent differences between synthetic and real data distributions. Smaller batch sizes tend to introduce more details/noises into the gradient updates during training, which, counterintuitively, can help in better generalizing to unseen data by avoiding overfitting to the synthetic dataset's general patterns. The noise can also act as a regularizer, preventing the model from becoming too confident in its predictions on the synthetic data, which may not fully capture the complexities of large batch-size data. Employing smaller batch sizes while training on the small-scale synthetic data allows models to explore the loss landscape more meticulously before converging to an optimal minimum. In Table 6, we empirically notice that utilizing a small batch size can improve model evaluation performance. This observed phenomenon aligns with the *Generalization Gap* theory, which arises when there is a lack of training samples.

### 3.3 Global-to-local Gradient Update via Curriculum

In SRe$^2$L (Yin et al., 2023) approach, the key of data synthesis revolves around utilizing the gradient information emanating from both the semantic class and the predictions of the pre-trained squeezing model, paired with BN distribution matching. Let $(\boldsymbol{x}, \boldsymbol{y})$ be an example $\boldsymbol{x}$ for optimization and its corresponding one-hot label $\boldsymbol{y}$ for the pre-trained squeezing model. Throughout the synthesis process, the squeezing model is frozen to recover the encoded information and ensure consistency and reliability in the generated data. Let $\mathcal{T}(\boldsymbol{x})$ be the target training distribution from which the data synthesis process should ultimately learn a function of desired trajectories, where $\mathcal{T}$ is a data transformation function to augment input samples to various levels of difficulties. Following Bengio et al. (2009), a weight $0 \le W_s(x) \le 1$ is defined and applied to example $\boldsymbol{x}$ at stage $s$ in the curriculum sequence. The training distribution $D_s(x)$ is:

$$D_s(x) \propto W_s(x)\mathcal{T}(x) \quad \forall x \tag{4}$$

In our scenario, since the varying difficulties are governed by the data transformation function $\mathcal{T}$, we can straightforwardly employ $W_s(x) = 1$ across all stages. Consequently, the training distribution solely depends on $\mathcal{T}(x)$ and can be simplified as follows:

$$D(x) \propto \mathcal{T}(x) \quad \forall x \tag{5}$$

By integrating curriculum learning within the data synthesis phase, this procedure can be defined as:

**Definition 1** (Curriculum Data Synthesis). *In the data synthesis optimization, the corresponding sequence of distributions $D(x)$ will be a curriculum if there is an increment in the entropy of these distributions, i.e., the difficulty of the transformed input samples escalates and becomes increasingly challenging for the pre-trained model to predict as the training progresses.*

Thus, the key for our curriculum data synthesis becomes how to design $\mathcal{T}(x)$ across different training iterations. The following section discusses several strategies to construct this in the curriculum scheme.

**Baseline: Constant Learning (CTL)**. This is the regular training method where all training examples are typically treated equally. Each sample from the training dataset has an equal chance of being transformed in a given batch, assuming no difficulty imbalance or biases across different training iterations.

CTL is straightforward to implement since we do not have to rank or organize examples based on difficulty. In practice, we use *RandomResizedCrop* to crop a small region via current crop ratio randomly sampled from a given interval [min_crop, max_crop] and then resize the cropped image to its original size, formulated as follows:

$$\boldsymbol{x}_{\mathcal{T}} \leftarrow RandomResizedCrop(\boldsymbol{x}_s, \texttt{min\_crop} = \alpha_l, \texttt{max\_crop} = \alpha_{\mathrm{u}}) \tag{6}$$

where $\alpha_l$ and $\alpha_{\mathrm{u}}$ are the constant lower and upper bounds of crop scale.

**Curriculum Learning (CL)**. As shown in Algorithm 1, in our CL, data samples are organized based on their difficulty. The difficulty level of the cropped region can be managed by defining the lower and upper scopes for the area ratio of the crop. This enables the assurance that specific crops of the image (small details or broader context) are included in the cropped region. For the difficulty adjustment, the rate at which more difficult examples are introduced and the criteria used to define difficulty are adjusted dynamically as predetermined using the following schedulers.

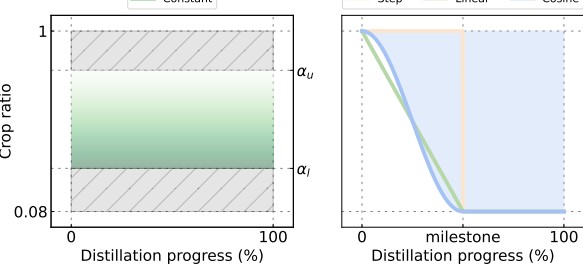

Figure 4: Illustration of global-to-local data synthesis. This figure shows our specific curriculum procedure in data synthesis to provide a comprehensive overview of our dataset distillation framework. It starts with a large area (single bounding-box in each step) to optimize the image, building a better initialization, and then gradually narrows down the image area of learning process so that it can focus on more detailed areas.

***Step***. Step scheduler reduces the minimal scale by a factor for every fixed or specified number of iterations, as shown in Figure 5 right.

***Linear***. Linear scheduler starts with a high initial value and decreases it linearly by a factor $\gamma$ to a minimum value over the whole training.

***Cosine***. Cosine scheduler modulates the distribution according to the cosine function of the current iteration number, yielding a smoother and more gradual adjustment compared to step-based methods.

As shown in Figure 5, the factor distribution manages the difficulty level of crops with adjustable $\alpha_u$ and $\alpha_l$ for CTL and milestone for CL.

Figure 5: Crop ratio schedulers of prior CTL solution (left) and our *Global-to-local* (right) enabled by curriculum. The colored regions depict the random sampling intervals for the crop ratio value in each iteration under different schedulers.

**Data Synthesis by Recovering and Relabeling.**

After receiving the transformed input $\boldsymbol{x}_{\mathcal{T}}$, we update it by aligning between the final classification label and intermediate Batch Normalization (BN) statistics, i.e., mean and variance from the original data. This stage forces the synthesized images to capture a shape of the original image distribution. The learning goal for this stage can be formulated as follows:

$$\boldsymbol{x}'_{\mathcal{T}} = \arg\min \ell\left(\phi_{\boldsymbol{\theta}}\left(\boldsymbol{x}_{\mathcal{T}}\right), \boldsymbol{y}\right) + \mathcal{R}_{\mathrm{reg}} \tag{7}$$

where $\phi_{\boldsymbol{\theta}}$ is the pre-trained squeezing model and will be frozen in this stage. During synthesis, only the input crop area will be updated by the gradient from the objective. The entire training procedure is illustrated in Figure 4. After synthesizing the data, we follow the relabeling process in SRe²L to generate soft labels using

---

**Algorithm 1:** Our CDA via *RandomResizedCrop*

---

**Input:** squeezed model $\phi_\theta$, recovery iteration $S$, curriculum milestone $T$, target label $\boldsymbol{y}$, default lower and upper bounds of crop scale $\beta_l$ and $\beta_\mathrm{u}$ in *RandomResizedCrop*, decay of lower scale bound $\gamma$

**Output:** synthetic image $\boldsymbol{x}$

**Initialize:** $x_0$ from a standard normal distribution

**for** step $s$ from 0 to $S$-1 **do**

   **if** $s \leq T$ **then**

$$\alpha \leftarrow \begin{cases} \beta_\mathrm{u} & \text{if step} \\ \beta_l + \gamma * (\beta_\mathrm{u} - s/T) & \text{if linear} \\ \beta_l + \gamma * (\beta_\mathrm{u} + \cos(\pi * s/T))/2 & \text{if cosine} \end{cases}$$

   **else**

     | $\alpha \leftarrow \beta_l$

   **end**

   $\boldsymbol{x}_\mathcal{T} \leftarrow RandomResizedCrop(\boldsymbol{x}_s, \mathtt{min\_crop} = \alpha, \mathtt{max\_crop} = \beta_\mathrm{u})$

   $\boldsymbol{x}'_\mathcal{T} \leftarrow \boldsymbol{x}_\mathcal{T}$ is optimized w.r.t $\phi_\theta$ and $\boldsymbol{y}$ in Eq. 7.

   $\boldsymbol{x}_{s+1} \leftarrow ReverseRandomResizedCrop(\boldsymbol{x}_s, \boldsymbol{x}'_\mathcal{T})$

**end**

**return** $\boldsymbol{x} \leftarrow \boldsymbol{x}_S$

---

FKD (Shen & Xing, 2022) with the integration of the small batch size setting for post-training. $\mathcal{R}_\mathrm{reg}$ is the regularization term used in Yin et al. (2023), its detailed formulation using channel-wise mean and variance matching is:

$$\begin{aligned} \mathcal{R}_\mathrm{reg}\left(\boldsymbol{x}'\right) &= \sum_k \left\| \mu_k\left(\boldsymbol{x}'\right) - \mathbb{E}\left(\mu_k \mid \mathcal{D}_o\right) \right\|_2 + \sum_k \left\| \sigma_l^2\left(\boldsymbol{x}'\right) - \mathbb{E}\left(\sigma_k^2 \mid \mathcal{D}_o\right) \right\|_2 \\ &\approx \sum_k \left\| \mu_k\left(\boldsymbol{x}'\right) - \mathbf{BN}_k^\mathrm{RM} \right\|_2 + \sum_k \left\| \sigma_k^2\left(\boldsymbol{x}'\right) - \mathbf{BN}_k^\mathrm{RV} \right\|_2 \end{aligned} \tag{8}$$

where $k$ is the index of BN layer, $\mu_k\left(\boldsymbol{x}'\right)$ and $\sigma_k^2\left(\boldsymbol{x}'\right)$ are the channel-wise mean and variance in current batch data. $\mathbf{BN}_k^\mathrm{RM}$ and $\mathbf{BN}_k^\mathrm{RV}$ are mean and variance in the pre-trained model at $k$-th BN layer, which are globally counted.

**Advantages of Global-to-local Synthesis**. The proposed `CDA` enjoys several advantages: **(1)** Stabilized training: Curriculum synthesis can provide a more stable training process as it reduces drastic loss fluctuations that can occur when the learning procedure encounters a challenging sample early on. **(2)** Better generalization: By gradually increasing the difficulty, the synthetic data can potentially achieve better generalization on diverse model architectures in post-training. It reduces the chance of the synthesis getting stuck in poor local minima early in the training process. **(3)** Avoid overfitting: By ensuring that the synthetic data is well-tuned on simpler examples before encountering outliers or more challenging data, there is a potential to reduce overfitting. Specifically, *better generalization* here refers to the ability of models trained on the distilled datasets to perform well across a wider range of evaluation scenarios. However, *avoiding overfitting* particularly refers to our curriculum strategy during the distillation process, where we use a flexible region update in each iteration to prevent overfitting that could occur with a fixed region update.

## 4 Experiments

### 4.1 Datasets and Implementation Details

We verify the effectiveness of our approach on small-scale CIFAR-100 and various ImageNet scale datasets, including Tiny-ImageNet (Le & Yang, 2015), ImageNet-1K (Deng et al., 2009), and ImageNet-21K (Ridnik et al., 2021). For evaluation, we train models from scratch on synthetic distilled datasets and report the Top-1 accuracy on real validation datasets. Default lower and upper bounds of crop scales $\beta_l$ and $\beta_u$ are 0.08 and 1.0, respectively. The decay $\gamma$ is 0.92. In Curriculum Learning (CL) settings, the actual lower bound is dynamically adjusted to control difficulty, whereas the upper bound is fixed to the default value of 1.0 to

Table 1: Comparison with state-of-the-art methods on various datasets.

| Dataset | CIFAR-100 | | Tiny-ImageNet | | | ImageNet-1K | | | | ImageNet-21K | |
|---|---|---|---|---|---|---|---|---|---|---|---|
| IPC | 10 | 50 | 10 | 50 | 100 | 10 | 50 | 100 | 200 | 10 | 20 |
| Ratio (%) | 2 | 10 | 2 | 10 | 20 | 0.8 | 4 | 8 | 16 | 0.8 | 1.6 |
| DM | 29.7±0.3 | 43.6±0.4 | 12.9±0.4 | 24.1±0.3 | - | 5.7±0.1 | 11.4±0.9 | - | - | - | - |
| DSA | 32.3±0.3 | 42.8±0.4 | - | - | - | - | - | - | - | - | - |
| FRePo | 42.5±0.2 | 44.3±0.2 | 25.4±0.2 | - | - | - | - | - | - | - | - |
| MTT | 39.7±0.4 | 47.7±0.2 | 23.2±0.2 | 28.0±0.3 | - | - | - | - | - | - | - |
| DataDAM | 34.8±0.5 | 49.4±0.3 | 18.7±0.3 | 28.7±0.3 | - | 6.3±0.0 | 15.5±0.2 | - | - | - | - |
| TESLA | 41.7±0.3 | 47.9±0.3 | - | - | - | 17.8±1.3 | 27.9±1.2 | - | - | - | - |
| DATM | 47.2±0.4 | 55.0±0.2 | **31.1±0.3** | 39.7±0.3 | - | - | - | - | - | - | - |
| *Full Dataset*[1] | 79.1 | | 61.2 | | | 69.8 | | | | 38.5 | |
| SRe²L | 23.5±0.8 | 51.4±0.8 | 17.7±0.7* | 41.1±0.4 | 49.7±0.3 | 21.3±0.6* | 46.8±0.2 | 52.8±0.4 | 57.0±0.3 | 18.5±0.2* | 21.8±0.1* |
| **CDA (Ours)** | **49.8±0.6** | **64.4±0.5** | 21.3±0.3 | **48.7±0.1** | **53.2±0.1** | **33.5±0.3** | **53.5±0.3** | **58.0±0.2** | **63.3±0.2** | **22.6±0.2** | **26.4±0.1** |

* Replicated experiment results are marked with *, while the other baseline results are referenced from original papers.
[1] The full dataset results refer to Top-1 val accuracy achieved by a ResNet-18 model trained on the full dataset, the architecture is the same as SRe²L and our CDA.

ensure there is a probability of cropping and optimizing the entire image in any progress. More details are provided in the Appendix B.

## 4.2 CIFAR-100

Result comparisons with baseline methods, including DM (Zhao & Bilen, 2023), DSA (Zhao & Bilen, 2021), FRePo (Zhou et al., 2022), MTT (Cazenavette et al., 2022b), DataDAM (Sajedi et al., 2023), TESLA (Cui et al., 2023), DATM (Guo et al., 2024), and

Table 2: Comparison on CIFAR-100.

| CIFAR-100 (IPC) | DC | DSA | DM | MTT | SRe²L | Ours |
|---|---|---|---|---|---|---|
| 1 | 12.8 | 13.9 | 11.4 | **24.3** | – | 13.4 |
| 10 | 25.2 | 32.3 | 29.7 | 40.1 | – | **49.8** |
| 50 | – | 42.8 | 43.6 | 47.7 | 49.4 | **64.4** |

SRe²L (Yin et al., 2023) on CIFAR-100 are presented in Table 2 and Table 1. Our model is trained with an 800ep budget. It can be observed that our CDA validation accuracy outperforms all baselines under 10 and 50 IPC. And our reported results have the potential to be further improved as training budgets increase. Overall, our CDA method is also applicable to small-scale dataset distillation.

## 4.3 Tiny-ImageNet

Results on the Tiny-ImageNet dataset are detailed in the second group of Table 1 and the first group of Table 4. Our CDA outperforms all baselines except DATM under 10 IPC. Compared to SRe²L, our CDA achieves average improvements of 7.7% and 3.4% under IPC 50 and IPC 100 settings across ResNet-{18, 50, 101} validation models, respectively. Importantly, CDA stands as the inaugural approach to diminish the Top-1 accuracy performance disparity to less than 10% between the distilled dataset employing IPC 100 and the full Tiny-ImageNet, signifying a breakthrough on this dataset.

## 4.4 ImageNet-1K

Table 3: Constant learning result. $\alpha_l$ and $\alpha_u$ stand for the `min_crop` and `max_crop` parameters in *RandomResizedCrop*. ‡ represents the results from SRe²L implementation but following the setting in the table.

| Constant learning type \ $\alpha$ | 0.08 | 0.2 | 0.4 | 0.6 | 0.8 | 1.0 |
|---|---|---|---|---|---|---|
| Easy ($\alpha_l = \alpha, \alpha_u = \beta_u$ (1.0)) | 44.90‡ | **47.88** | 46.34 | 45.35 | 43.48 | 41.30 |
| Hard ($\alpha_l = \beta_l$ (0.08), $\alpha_u = \alpha$) | 22.99 | 34.75 | 42.76 | 44.61 | **45.76** | 44.90‡ |

**Constant Learning (CTL)**. We leverage a ResNet-18 and employ synthesized data with 1K recovery iterations. As observed in Table 3, the results for exceedingly straightforward or challenging scenarios fall below the reproduced SRe²L baseline accuracy of 44.90%, especially when $\alpha \geq 0.8$ in *easy* and $\alpha \leq 0.4$ in *hard* type. Thus, the results presented in Table 3 suggest that adopting a larger cropped range assists in circumventing extreme scenarios, whether easy or hard, culminating in enhanced performance. A noteworthy

Table 4: Comparison with baseline on various datasets.

| Dataset | IPC | ResNet-18 | | ResNet-50 | | ResNet-101 | |
|---------|-----|-----------|------|-----------|------|------------|------|
| | | $SRe^2L$ | Ours | $SRe^2L$ | Ours | $SRe^2L$ | Ours |
| Tiny-IN | 50 | 41.1 | $48.7^{\uparrow 7.6}$ | 42.2 | $49.7^{\uparrow 7.5}$ | 42.5 | $50.6^{\uparrow 8.1}$ |
| | 100 | 49.7 | $53.2^{\uparrow 3.5}$ | 51.2 | $54.4^{\uparrow 3.2}$ | 51.5 | $55.0^{\uparrow 3.5}$ |
| IN-1K | 50 | 46.8 | $53.5^{\uparrow 6.7}$ | 55.6 | $61.3^{\uparrow 5.7}$ | 57.6 | $61.6^{\uparrow 4.0}$ |
| | 100 | 52.8 | $58.0^{\uparrow 5.2}$ | 61.0 | $65.1^{\uparrow 4.1}$ | 62.8 | $65.9^{\uparrow 3.1}$ |
| | 200 | 57.0 | $63.3^{\uparrow 6.3}$ | 64.6 | $67.6^{\uparrow 3.0}$ | 65.9 | $68.4^{\uparrow 2.5}$ |
| IN-21K | 10 | 18.5 | $22.6^{\uparrow 4.1}$ | 27.4 | $32.4^{\uparrow 5.0}$ | 27.3 | $34.2^{\uparrow 6.9}$ |
| | 20 | 21.8 | $26.4^{\uparrow 4.6}$ | 31.3 | $35.3^{\uparrow 4.0}$ | 33.2 | $36.1^{\uparrow 2.9}$ |

observation is the crucial role of appropriate lower and upper bounds for constant learning in boosting validation accuracy. This highlights the importance of employing curriculum data augmentation strategies in data synthesis.

**Curriculum Learning (CL).** We follow the recovery recipe of $SRe^2L$'s best result for 4K recovery iterations. As illustrated in Table 1 and the second group of Table 4, when compared to the strong baseline $SRe^2L$, CDA enhances the validation accuracy, exhibiting average margins of 6.1%, 4.3%, and 3.2% on ResNet-{18, 50, 101} across varying IPC settings. Furthermore, as shown in Figure 1, the results achieve with our CDA utilizing merely 1K recovery iterations surpass those of $SRe^2L$ encompassing the entire 4K iterations. These results substantiate the efficacy and effectiveness of applying CDA in large-scale dataset distillation.

## 4.5 ImageNet-21K

**Pre-training Results.** Table 21 of the Appendix presents the accuracy for ResNet-18 and ResNet-50 on ImageNet-21K-P, considering varying initial weight configurations. Models pre-trained by us and initialized with ImageNet-1K weight exhibit commendable accuracy, showing a 2.0% improvement, while models initialized randomly achieve marginally superior accuracy. We utilize these pre-trained models to recover ImageNet-21K data and to assign labels to the synthetic images generated. An intriguing observation is the heightened difficulty in data recovering from pre-trained models that are initialized randomly compared to those initialized with ImageNet-1K weight. Thus, our experiments employ CDA specifically on pre-trained models that are initialized with ImageNet-1K weight.

**Validation Results.** As illustrated in Table 1 and the final group of Table 4, we perform validation experiments on the distilled ImageNet-21K employing IPC 10 and 20. This yields an extreme compression ratio of 100× and 50×. When applying IPC 10, i.e., the models are trained utilizing a distilled dataset that is a mere 1% of the full dataset. Remarkably, validation accuracy surpasses 20% and 30% on ResNet-18 and ResNet-{50, 101}, respectively. Compared to reproduced $SRe^2L$ on ImageNet-21K, our approach attains an elevation of 5.3% on average under IPC 10/20. This not only highlights the efficacy of our approach in maintaining dataset essence despite high compression but also showcases the potential advancements in accuracy over existing methods.

## 4.6 Ablations

Table 5: Applicability of our proposed method on SC-DD (Zhou et al., 2024a).

| Method | Tiny-ImageNet | ImageNet-1K |
|--------|---------------|-------------|
| SC-DD | 45.5 | 53.1 |
| SC-DD+Curriculum (Ours) | $46.5^{\uparrow 1.0}$ | $54.0^{\uparrow 0.9}$ |

**Applicability.** Our method is designed to be applicable to general decoupled dataset distillation approaches. To demonstrate this, we apply our approach to the self-supervised SC-DD (Zhou et al., 2024b). As shown in Table 5, our method achieves 1.0 and 0.9 improvements on Tiny-ImageNet and ImageNet-1K, respectively.

**Curriculum Scheduler.** To schedule the global-to-local learning, we present three distinct types of curriculum schedulers, *step*, *linear*, and *cosine* to manipulate the lower bounds on data cropped augmentation. As illustrated in Figure 5, the dataset distillation progress is divided into two phases by a milestone. It is observed that both *linear* and *cosine* with continuous decay manifest robustness across diverse milestone configurations and reveal a trend of enhancing accuracy performance when the milestone is met at a later phase, as shown in Figure 6. Moreover, *cosine* marginally outperforms *linear* in terms of accuracy towards the end. Consequently, we choose to implement the *cosine* scheduler, assigning a milestone percentage of 1.0, to modulate the minimum crop ratio adhering to the principles of curriculum learning throughout the progression of synthesis.

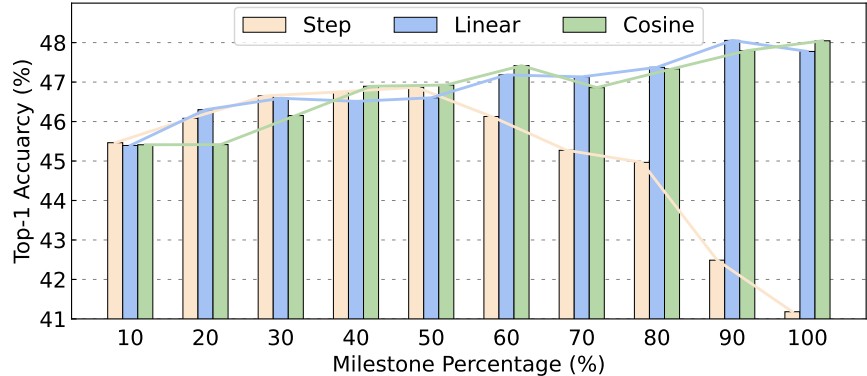

Figure 6: Ablation study on three different schedulers with varied milestone settings. Each number is obtained by averaging repeated experiments with three different seeds. The figure shows that the difference between linear and cosine schedulers is marginal and the best result for linear is at the milestone of 90% while the cosine scheduler performs similarly or better in the end. To avoid manually setting the milestone percentage for the linear scheduler, we adopt the cosine scheduler with a milestone percentage of 100% in our experiments.

**Batch Size in Post-training.** We perform an ablation study to assess the influence of utilizing smaller batch sizes on the generalization performance of models when the synthetic data is limited. We report results on the distilled ImageNet-21K from ResNet-18. In Table 6, a rise in validation accuracy is observed as batch size reduces, peaking at 16. This suggests that smaller batch sizes enhance performance on small-scale synthetic datasets. However, this leads to more frequent data loading and lower GPU utilization in our case, extending training times. To balance training time with performance, we chose a batch size of 32 for our experiments.

### 4.7 Analysis

**Cross-Model Generalization.** The challenge of ensuring distilled datasets generalize effectively across models unseen during the recovery phase remains significant, as in prior approaches (Zhao et al., 2020; Cazenavette et al., 2022a), synthetic images were optimized to overfit the recovery model. In the first group of Table 7, we deploy our ImageNet-1K distilled datasets to train various validation models, and we attain over 60% Top-1 accuracy with most of these models. Additionally, our performance in Top-1 accuracy surpasses that of SRe$^2$L across all validation models spanning various architectures. Although DeiT-Tiny is not validated with a comparable Top-1 accuracy to other CNN models due to the ViT's inherent characteristic requiring more training data, CDA achieves double cross-model generation performance on the DeiT-Tiny validation model, compared with SRe$^2$L.

Table 6: Ablation on batch size in validation.

| Batch Size | Acc. (%) |
|---|---|
| 128 | 20.79 |
| 64 | 21.85 |
| 32 | 22.54 |
| 16 | **22.75** |
| 8 | 22.41 |

More validation models on distilled ImageNet-1K are included in Table 19 of the Appendix. The second group of Table 7 supports further empirical substantiation of the CDA's efficacy in the distillation of large-scale ImageNet-21K datasets. The results demonstrate that the CDA's distilled datasets exhibit reduced dependency on specific recovery models, thereby further alleviating the overfitting optimization issues.

Table 7: Cross-model generation on distilled ImageNet-1K with 50 IPC and ImageNet-21K with 20 IPC.

| Dataset | Method | Validation Model | | | | | | |
|---------|--------|------|------|------|-------------|--------------|--------------|-----------|
| | | R18 | R50 | R101 | DenseNet-121 | RegNet-Y-8GF | ConvNeXt-Tiny | DeiT-Tiny |
| IN-1K | SRe$^2$L | 46.80 | 55.60 | 57.60 | 49.74 | 60.34 | 53.53 | 15.41 |
| | CDA (ours) | **53.45** | **61.26** | **61.57** | **57.35** | **63.22** | **62.58** | **31.95** |
| IN-21K | SRe$^2$L | 21.83 | 31.26 | 33.24 | 24.66 | 34.22 | 34.95 | 15.76 |
| | CDA (ours) | **26.42** | **35.32** | **36.12** | **28.66** | **36.13** | **36.31** | **18.56** |

**Impact of Curriculum**. To study the curriculum's advantage on synthetic image characteristics, we evaluate the Top-1 accuracy on CDA, SRe$^2$L and real ImageNet-1K training set, using a mean of random 10-crop and global images. We employ PyTorch's pre-trained MobileNet-V2 to classify these images. As shown in Table 8, CDA images closely resemble real ImageNet images in prediction accuracies, better than SRe$^2$L. Consequently, curriculum data augmentation improves global image prediction and reduces bias and overfitting post-training on simpler, cropped images of SRe$^2$L.

Table 8: Classification accuracy using MobileNet-V2.

| Top-1 (%) | Dataset | | |
|-----------|---------|------------|-------|
| | SRe$^2$L | CDA (ours) | Real |
| global | 79.34 | 81.25 | 82.16 |
| cropped | 87.48 | 82.44 | 72.73 |

**Visualization and Discussion**. Figure 7 provides a comparative visualization of the gradient synthetic images at recovery steps of {100, 500, 1K, 2K} to illustrate the differences between SRe$^2$L and CDA within the dataset distillation process. SRe$^2$L images in the upper line exhibit a significant amount of noise, indicating a slow recovery progression in the early recovery stage. On the contrary, due to the mostly entire image optimization in the early stage, CDA images in the lower line can establish the layout of the entire image and reduce noise rapidly. And the final synthetic images contain more visual information directly related to the target class *Plant*. Therefore, the comparison highlights CDA's ability to synthesize images with enhanced visual coherence to the target class, offering a more efficient recovery process. More visualizations are provided in Appendix D.

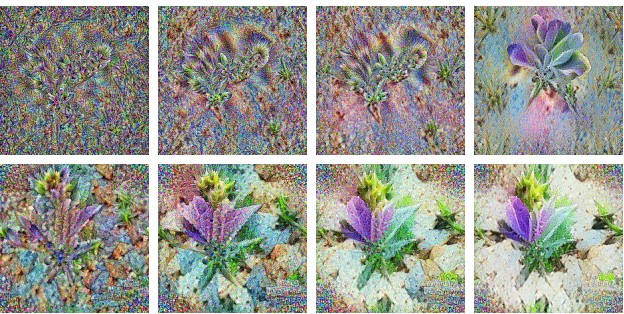

Figure 7: Synthetic ImageNet-21K images (*Plant*).

**Synthesis Cost**. We highlight that there is no additional synthesis cost incurred in our CDA to SRe$^2$L (Yin et al., 2023) under the same recovery iteration setting. Specifically, for ImageNet-1K, it takes about 29 hours to generate the distilled ImageNet-1K with 50 IPC on a single A100 (40G) GPU and the peak GPU memory utilization is 6.7GB. For ImageNet-21K, it takes 11 hours to generate ImageNet-21K images per IPC on a single RTX 4090 GPU and the peak GPU memory utilization is 15GB. In our experiment, it takes about 55 hours to generate the entire distilled ImageNet-21K with 20 IPC on 4× RTX 4090 GPUs in total. Selecting representative real images for input data initialization instead of the Gaussian noise initialization will be an effective way to further accelerate image distillation and reduce synthesis costs. It can potentially reduce the number of recovery iterations required to achieve high-quality distilled data and this approach could lead to faster convergence and lower synthesis costs. Nevertheless, we present our detailed training time on several tested models, as shown in Table 9.

Table 9: Detailed training time on different models.

| Model | ResNet-18 | ResNet-50 | ResNet-101 | DenseNet-121 | RegNet-Y-8GF | ConvNeXt-Tiny | DeiT-Tiny |
|-------|-----------|-----------|------------|--------------|--------------|---------------|-----------|
| Training time (A100 GPU hours) | 2.3 | 7.1 | 12.5 | 9.1 | 13.6 | 18.3 | 4.6 |

## 4.8 Application: Continual Learning

The distilled datasets, comprising high-semantic images, possess a boosted representation capacity compared to the original datasets. This attribute can be strategically harnessed to combat catastrophic forgetting in continual learning. We have further validated the effectiveness of our introduced CDA synthesis within

various continual learning scenarios. Following the setting introduced in $SRe^2L$ (Yin et al., 2023), we conducted 5-step and 10-step class-incremental experiments on Tiny-ImageNet, aligning our results against the baseline $SRe^2L$ and a randomly selected subset on Tiny-ImageNet for comparative analysis. As illustrated in Figure 8, our `CDA` distilled dataset notably surpasses $SRe^2L$, exhibiting an average advantage of 3.8% and 4.5% on 5-step and 10-step class-incremental learning assignments respectively. This demonstrates the substantial benefits inherent in the generation of `CDA`, particularly in mitigating the complexities associated with continual learning.

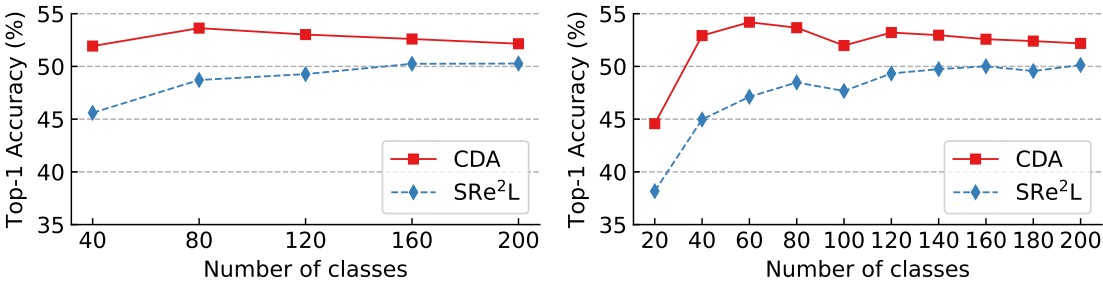

Figure 8: 5/10-step class-incremental learning on Tiny-IN.

## 5   Conclusion

We presented a new framework focused on *global-to-local* gradient refinement through curriculum data synthesis for large-scale dataset distillation. Our approach involves a practical paradigm with detailed pertaining for compressing knowledge, data synthesis for recovery, and post-training recipes. The proposed approach enables the distillation of ImageNet-21K to 50× smaller while maintaining competitive accuracy levels. In regular benchmarks, such as ImageNet-1K and CIFAR-100, our approach also demonstrated superior performance, surpassing prior state-of-the-art methods by substantial margins. We further show the capability of our synthetic data on downstream tasks of cross-model generalization and continual learning. With the recent substantial growth in the size of both models and datasets, the critical need for dataset distillation on large-scale datasets and models has become increasingly prominent and urgent. Our future work will focus on distilling more modalities like language and speech.

**Limitations.** Our proposed approach is robust to generate informative images, while we also clarify that the quality of our generated data is not comparable to the image quality achieved by state-of-the-art generative models on large-scale datasets. This difference is expected, given the distinct goals of dataset distillation versus generative models. Generative models aim to synthesize highly realistic images with detailed features, whereas dataset distillation methods focus on producing images that capture the most representative information possible for efficient learning in downstream tasks. Realism is not the primary objective in dataset distillation.

### Acknowledgments

This research is supported by the MBZUAI-WIS Joint Program for AI Research and the Google Research award grant.

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

## Appendix

## A    Datasets Details

We conduct experiments on three ImageNet scale datasets, Tiny-ImageNet (Le & Yang, 2015), ImageNet-1K (Deng et al., 2009), and ImageNet-21K (Ridnik et al., 2021). The dataset details are as follows:

- CIFAR-100 dataset composes 500 training images per class, each with a resolution of 32×32 pixels, across 100 classes.

- Tiny-ImageNet dataset is derived from ImageNet-1K and consists of 200 classes. Within each category, there are 500 images with a uniform 64×64 resolution.

- ImageNet-1K dataset comprises 1,000 classes and 1,281,167 images in total. We resize all images into standard 224×224 resolution during the data loading stage.

- The original ImageNet-21K dataset is an extensive visual recognition dataset containing 21,841 classes and 14,197,122 images. We use ImageNet-21K-P (Ridnik et al., 2021) which utilizes data processing to remove infrequent classes and resize all images to 224×224 resolution. After data processing, ImageNet-21K-P dataset consists of 10,450 classes and 11,060,223 images.

## B    Implementation Details

### B.1    CIFAR-100

**Hyper-parameter Setting.** We train a modified ResNet-18 model (He et al., 2020) on CIFAR-100 training data with a Top-1 accuracy of 79.1% using the parameter setting in Table 10a. The well-trained model serves as the recovery model under the recovery setting in Table 10b.

Table 10: Hyper-parameter settings on CIFAR-100.

(a) Squeezing/validation setting.

| config | value |
| --- | --- |
| optimizer | SGD |
| base learning rate | 0.1 |
| momentum | 0.9 |
| weight decay | 5e-4 |
| batch size | 128 (squeeze) / 8 (val) |
| learning rate schedule | cosine decay |
| training epoch | 200 (squeeze) / 800 (val) |
| augmentation | RandomResizedCrop |

(b) Recovery setting.

| config | value |
| --- | --- |
| $\alpha_{BN}$ | 0.01 |
| optimizer | Adam |
| base learning rate | 0.25 |
| momentum | $\beta_1$, $\beta_2 = 0.5$, 0.9 |
| batch size | 100 |
| learning rate schedule | cosine decay |
| recovery iteration | 1,000 |
| augmentation | RandomResizedCrop |

Due to the low resolution of CIFAR images, the default lower bound $\beta_l$ needs to be raised from 0.08 (ImageNet setting) to a higher reasonable value in order to avoid the training inefficiency caused by extremely small cropped areas with little information. Thus, we conducted the ablation to select the optimal value for the default lower bound $\beta_l$ in RandomResizedCrop operations in Table 11. We choose 0.4 as the default lower bound $\beta_l$ in Algorithm 1 to exhibit the best distillation performance on CIFAR-100. We adopt a small batch size value of 8 and extend the training budgets in the following validation stage, which aligns with the strong training recipe on inadequate datasets.

Table 11: Ablation on the lower bound $\beta_l$ setting in distilling CIFAR-100.

| default lower bound $\beta_l$ | 0.08 | 0.2 | 0.4 | 0.6 | 0.8 | 1.0 |
| --- | --- | --- | --- | --- | --- | --- |
| validation accuracy (800ep) (%) | 58.5 | 62.14 | **64.0** | 63.36 | 61.65 | 54.43 |

## B.2 Tiny-ImageNet

**Hyper-parameter Setting.** We train a modified ResNet-18 model (He et al., 2020) on Tiny-ImageNet training data with the parameter setting in Table 12a and use the well-trained ResNet-18 model with a Top-1 accuracy of 61.2% as a recovery model for `CDA`. The recovery setting is provided in Table 12b.

Table 12: Hyper-parameter settings on Tiny-ImageNet.



(a) Squeezing/validation setting.

| config | value |
|---|---|
| optimizer | SGD |
| base learning rate | 0.2 |
| momentum | 0.9 |
| weight decay | 1e-4 |
| batch size | 256 (squeeze) / 64 (val) |
| learning rate schedule | cosine decay |
| training epoch | 50 (squeeze) / 100 (val) |
| augmentation | RandomResizedCrop |

(b) Recovery setting.

| config | value |
|---|---|
| $\alpha_{\mathrm{BN}}$ | 1.0 |
| optimizer | Adam |
| base learning rate | 0.1 |
| momentum | $\beta_1$, $\beta_2 = 0.5$, 0.9 |
| batch size | 100 |
| learning rate schedule | cosine decay |
| recovery iteration | 4,000 |
| augmentation | RandomResizedCrop |



**Small IPC Setting Comparison.** Table 13 presents the result comparison among our `CDA`, DM (Zhao & Bilen, 2023) and MTT (Cazenavette et al., 2022b). Consider that our approach is a decoupled process of dataset compression followed by recovery through gradient updating. It is well-suited to large-scale datasets but less so for small IPC values. As anticipated, there is no advantage when IPC value is extremely low, such as IPC = 1. However, when the IPC is increased slightly, our method demonstrates considerable benefits on accuracy over other counterparts. Furthermore, we emphasize that our approach yields substantial improvements when afforded a larger training budget, i.e., more training epochs.

Table 13: Comparison with baseline methods on Tiny-ImageNet.

| Tiny-ImageNet IPC | DM | MTT | CDA (200ep) | CDA (400ep) | CDA (800ep) |
|---|---|---|---|---|---|
| 1 | 3.9 | **8.8** | $2.38 \pm 0.08$ | $2.82 \pm 0.06$ | $3.29 \pm 0.26$ |
| 10 | 12.9 | 23.2 | $30.41 \pm 1.53$ | $37.41 \pm 0.02$ | $\mathbf{43.04 \pm 0.26}$ |
| 20 | – | – | $43.93 \pm 0.20$ | $47.76 \pm 0.19$ | $\mathbf{50.46 \pm 0.14}$ |
| 50 | 24.1 | 28.0 | $50.26 \pm 0.09$ | $51.52 \pm 0.17$ | $\mathbf{55.50 \pm 0.18}$ |

**Continual Learning.** We adhere to the continual learning codebase outlined in Zhao et al. (2020) and validate provided SRe$^2$L and our `CDA` distilled Tiny-ImageNet dataset under IPC 100 as illustrated in Figure 8. Detailed values are presented in the Table 14 and Table 15.

Table 14: 5-step class-incremental learning on Tiny-ImageNet. This complements details in the left subfigure of Figure 8.

| # class | 40 | 80 | 120 | 160 | 200 |
|---|---|---|---|---|---|
| SRe$^2$L | 45.60 | 48.71 | 49.27 | 50.25 | 50.27 |
| CDA (ours) | 51.93 | 53.63 | 53.02 | 52.60 | 52.15 |

Table 15: 10-step class-incremental learning on Tiny-ImageNet. This complements details in the right subfigure of Figure 8.

| # class | 20 | 40 | 60 | 80 | 100 | 120 | 140 | 160 | 180 | 200 |
|---|---|---|---|---|---|---|---|---|---|---|
| SRe$^2$L | 38.17 | 44.97 | 47.12 | 48.48 | 47.67 | 49.33 | 49.74 | 50.01 | 49.56 | 50.13 |
| CDA (ours) | 44.57 | 52.92 | 54.19 | 53.67 | 51.98 | 53.21 | 52.96 | 52.58 | 52.40 | 52.18 |

### B.3 ImageNet-1K

**Hyper-parameter Settings.** We employ PyTorch off-the-shelf ResNet-18 and DenseNet-121 with the Top-1 accuracy of {69.8%, 74.4%} which are trained with the official recipe in Table 16a. And the recovery settings are provided in Table 16c, and it is noteworthy that we tune and set distinct parameters $\alpha_{\text{BN}}$ and learning rate for different recovery models in Table 16d. Then, we employ ResNet-{18, 50, 101, 152} (He et al., 2016), DenseNet-121 (Huang et al., 2017), RegNet (Radosavovic et al., 2020), ConvNeXt (Liu et al., 2022b), and DeiT-Tiny (Touvron et al., 2021) as validation models to evaluate the cross-model generalization on distilled ImageNet-1K dataset under the validation setting in Table 16b.

Table 16: Hyper-parameter settings on ImageNet-1K.

(a) Squeezing setting.

| config | value |
| --- | --- |
| optimizer | SGD |
| base learning rate | 0.1 |
| momentum | 0.9 |
| weight decay | 1e-4 |
| batch size | 256 |
| lr step size | 30 |
| lr gamma | 0.1 |
| training epoch | 90 |
| augmentation | RandomResizedCrop |

(b) Validation setting.

| config | value |
| --- | --- |
| optimizer | AdamW |
| base learning rate | 1e-3 |
| weight decay | 1e-2 |
| batch size | 128 |
| learning rate schedule | cosine decay |
| training epoch | 300 |
| augmentation | RandomResizedCrop |

(c) Shared recovery setting.

| config | value |
| --- | --- |
| optimizer | Adam |
| momentum | $\beta_1, \beta_2 = 0.5, 0.9$ |
| batch size | 100 |
| learning rate schedule | cosine decay |
| augmentation | RandomResizedCrop |

(d) Model-specific recovery setting.

| config | ResNet-18 | DenseNet-121 |
| --- | --- | --- |
| $\alpha_{\text{BN}}$ | 0.01 | 0.01 |
| base learning rate | 0.25 | 0.5 |
| recovery iteration | 1,000 / 4,000 | 1,000 |

**Histogram Values.** The histogram data of ImageNet-1K comparison with SRe2L in Figure 1 can be conveniently found in the following Table 17 for reference.

Table 17: ImageNet-1K comparison with SRe$^2$L. This table complements details in Figure 1.

| Method \ Validation Model | ResNet-18 | ResNet-50 | ResNet-101 | DenseNet-121 | RegNet-Y-8GF |
| --- | --- | --- | --- | --- | --- |
| SRe$^2$L (4K) | 46.80 | 55.60 | 57.59 | 49.74 | 60.34 |
| Our CDA (1K) | 52.88 | 60.70 | 61.10 | 57.26 | 62.94 |
| Our CDA (4K) | 53.45 | 61.26 | 61.57 | 57.35 | 63.22 |

To conduct the ablation studies efficiently in Table 3, Table 22 and Figure 6, we recover the data for 1,000 iterations and validate the distilled dataset with a batch size of 1,024, keeping other settings the same as Table 16. Detailed values of the ablation study on schedulers are provided in Table 18.

Table 18: Ablation study on three different schedulers with varied milestone settings. It complements details in Figure 6.

| Scheduler \ Milestone | 0.1 | 0.2 | 0.3 | 0.4 | 0.5 | 0.6 | 0.7 | 0.8 | 0.9 | 1 |
|---|---|---|---|---|---|---|---|---|---|---|
| Step | 45.46 | 46.08 | 46.65 | 46.75 | 46.87 | 46.13 | 45.27 | 44.97 | 42.49 | 41.18 |
| Linear | 45.39 | 46.30 | 46.59 | 46.51 | 46.60 | 47.18 | 47.13 | 47.37 | 48.06 | 47.78 |
| Cosine | 45.41 | 45.42 | 46.15 | 46.90 | 46.93 | 47.42 | 46.86 | 47.33 | 47.80 | 48.05 |

**Cross-Model Generalization.** To supplement the validation models on distilled ImageNet-1K in Table 7, including more different architecture models to evaluate the cross-architecture performance. We have conducted validation experiments on a broad range of models, including SqueezeNet, MobileNet, EfficientNet, MNASNet, ShuffleNet, ResMLP, AlexNet, DeiT-Base, and VGG family models. These validation models are selected from a wide variety of architectures, encompassing a vast range of parameters, shown in Table 19. In the upper group of the table, the selected models are relatively small and efficient. There is a trend that its validation performance improves as the number of model parameters increases. In the lower group, we validated earlier models AlexNet and VGG. These models also show a trend of performance improvement with increasing size, but due to the simplicity of early model architectures, such as the absence of residual connections, their performance is inferior compared to more recent models. Additionally, we evaluated our distilled dataset on ResMLP, which is based on MLPs, and the DeiT-Base model, which is based on transformers. In summary, the distilled dataset created using our `CDA` method demonstrates strong validation performance across a wide range of models, considering both architectural diversity and parameter size.

Table 19: ImageNet-1K Top-1 on cross-model generation. Our `CDA` dataset consists of 50 IPC.

| Model | SqueezeNet | MobileNet | EfficientNet | MNASNet | ShuffleNet | ResMLP |
|---|---|---|---|---|---|---|
| #Params (M) | 1.2 | 3.5 | 5.3 | 6.3 | 7.4 | 30.0 |
| accuracy (%) | 19.70 | 49.76 | 55.10 | 55.66 | 54.69 | 54.18 |
| Model | AlexNet | DeiT-Base | VGG-11 | VGG-13 | VGG-16 | VGG-19 |
| #Params (M) | 61.1 | 86.6 | 132.9 | 133.0 | 138.4 | 143.7 |
| accuracy (%) | 14.60 | 30.27 | 36.99 | 38.60 | 42.28 | 43.30 |

### B.4 ImageNet-21K

**Hyper-parameter Setting.** ImageNet-21K-P (Ridnik et al., 2021) proposes two training recipes to train ResNet-{18, 50} models. One way is to initialize the models from well-trained ImageNet-1K weight and train on ImageNet-21K-P for 80 epochs, another is to train models with random initialization for 140 epochs, as shown in Table 20a. The accuracy metrics on both training recipes are reported in Table 21. In our experiments, we utilize the pre-trained ResNet-{18, 50} models initialized by ImageNet-1K weight with the Top-1 accuracy of {38.1%, 44.2%} as recovery model. And the recovery setting is provided in Table 20c. Then, we evaluate the quality of the distilled ImageNet-21K dataset on ResNet-{18, 50, 101} validation models under the validation setting in Table 20b. To accelerate the ablation study on the batch size setting in Table 6, we train the validation model ResNet-18 for 140 epochs.

## C  Reverse Curriculum Learning

**Reverse Curriculum Learning (RCL).** We use a reverse step scheduler in the RCL experiments, starting with the default cropped range from $\beta_l$ to $\beta_u$ and transitioning at the milestone point to optimize the whole image, shifting from challenging to simpler optimizations. Other settings follow the recovery recipe on ResNet-18 for 1K recovery iterations. Table 22 shows the RCL results, a smaller step milestone indicates an earlier difficulty transition. The findings reveal that CRL does not improve the generated dataset's quality compared to the baseline SRe$^2$L, which has 44.90% accuracy.

Table 20: Hyper-parameter settings on ImageNet-21K.

(a) Squeezing setting.

| config | value |
|---|---|
| optimizer | Adam |
| base learning rate | 3e-4 |
| weight decay | 1e-4 |
| batch size | 1,024 |
| learning rate schedule | cosine decay |
| label smooth | 0.2 |
| training epoch | 80/140 |
| augmentation | CutoutPIL, RandAugment |

(b) Validation setting.

| config | value |
|---|---|
| optimizer | AdamW |
| base learning rate | 2e-3 |
| weight decay | 1e-2 |
| batch size | 32 |
| learning rate schedule | cosine decay |
| label smooth | 0.2 |
| training epoch | 300 |
| augmentation | CutoutPIL, RandomResizedCrop |

(c) Recovery setting.

| config | value |
|---|---|
| $\alpha_{BN}$ | 0.25 |
| optimizer | Adam |
| base learning rate | 0.05 (ResNet-18), 0.1 (ResNet-50) |
| momentum | $\beta_1$, $\beta_2 = 0.5$, 0.9 |
| batch size | 100 |
| learning rate schedule | cosine decay |
| recovery iteration | 2,000 |
| augmentation | RandomResizedCrop |

Table 21: Accuracy of ResNet-{18, 50} on ImageNet-21K-P.

| Model | Initial Weight | Top-1 Acc. (%) | Top-5 Acc. (%) |
|---|---|---|---|
| ResNet-18 (Ours) | ImageNet-1K | 38.1 | 67.2 |
| | Random | 38.5 | 67.8 |
| Ridnik et al. (2021) | ImageNet-1K | 42.2 | 72.0 |
| ResNet-50 (Ours) | ImageNet-1K | 44.2$^{\uparrow 2.0}$ | 74.6$^{\uparrow 2.6}$ |
| | Random | 44.5$^{\uparrow 2.3}$ | 75.1$^{\uparrow 3.1}$ |

Table 22: Ablation of reverse curriculum learning.

| Step Milestone | Accuracy (%) |
|---|---|
| 0.2 | 41.38 |
| 0.4 | 41.59 |
| 0.6 | 42.60 |
| 0.8 | 44.39 |

## D   Visulization

We provide additional comparisons of four groups of visualizations on synthetic ImageNet-21K images at recovery steps of {100, 500, 1,000, 1,500, 2,000} between SRe$^2$L (upper) and `CDA` (lower) in Figure 9. The chosen target classes are *Benthos*, *Squash Rackets*, *Marine Animal*, and *Scavenger*.

In addition, we present our `CDA`'s synthetic ImageNet-1K images in Figure 10 and ImageNet-21K images in Figure 11 and Figure 12.

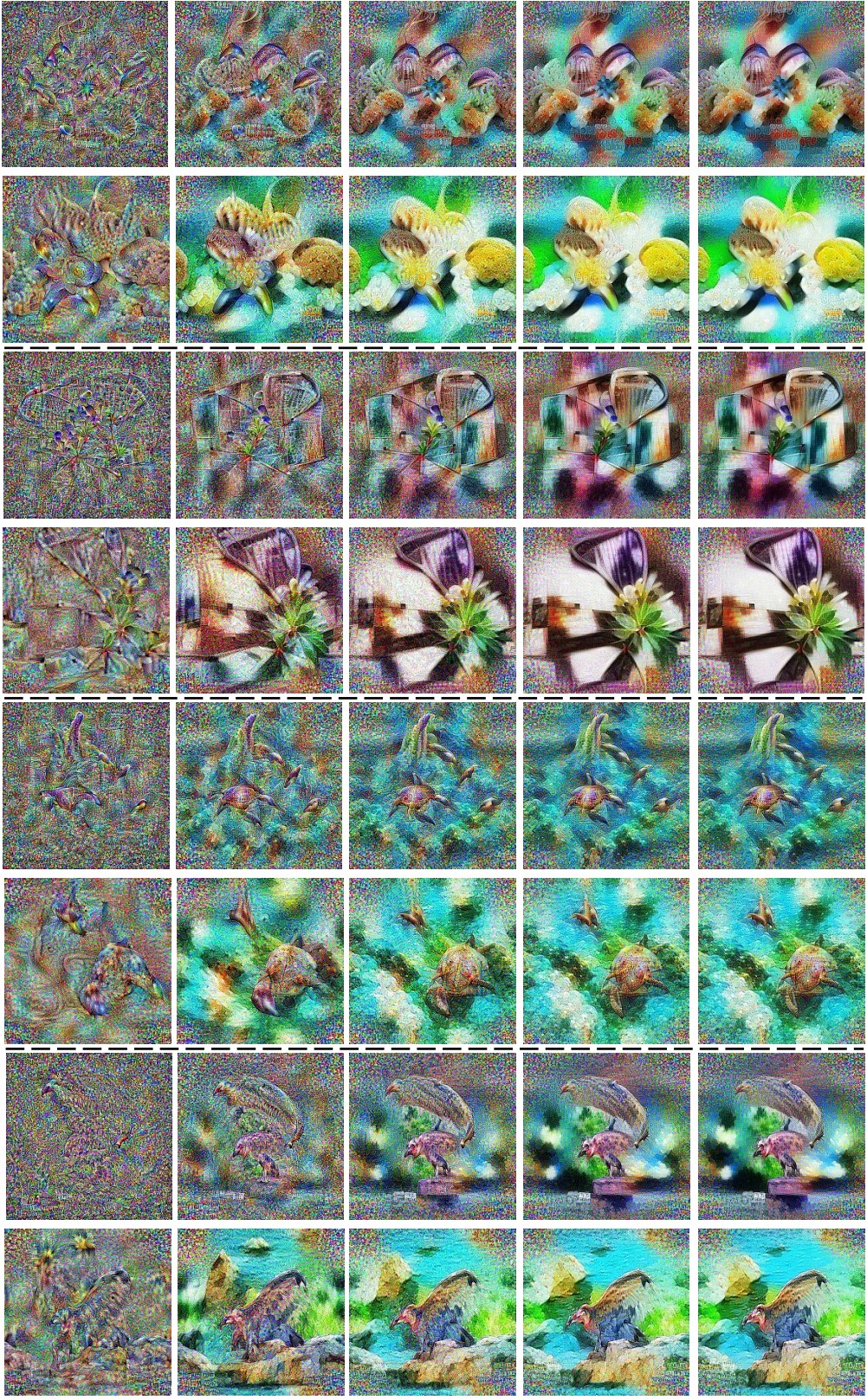

Figure 9: Synthetic ImageNet-21K data visualization comparison.

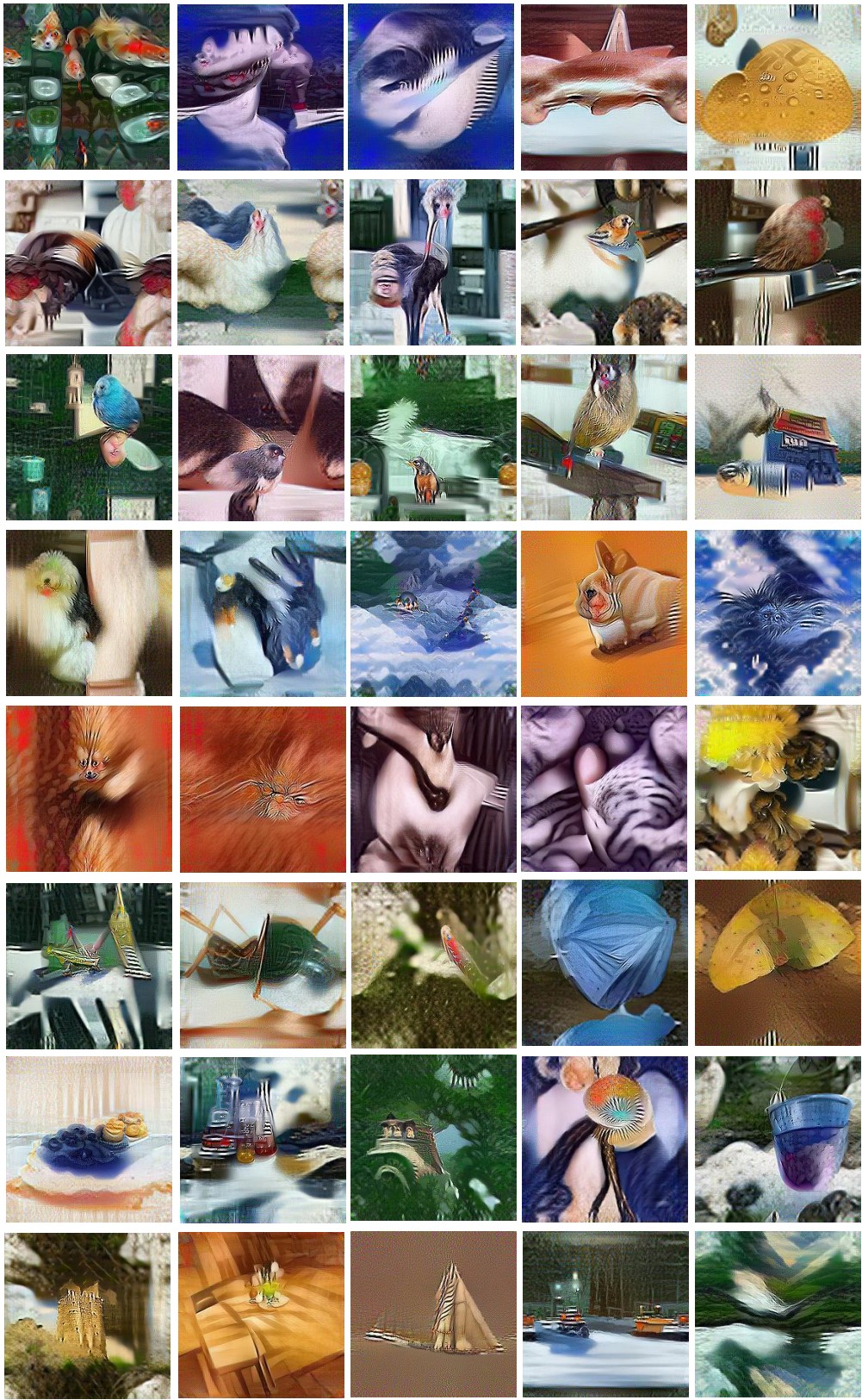

Figure 10: Synthetic ImageNet-1K data visualization from `CDA`.

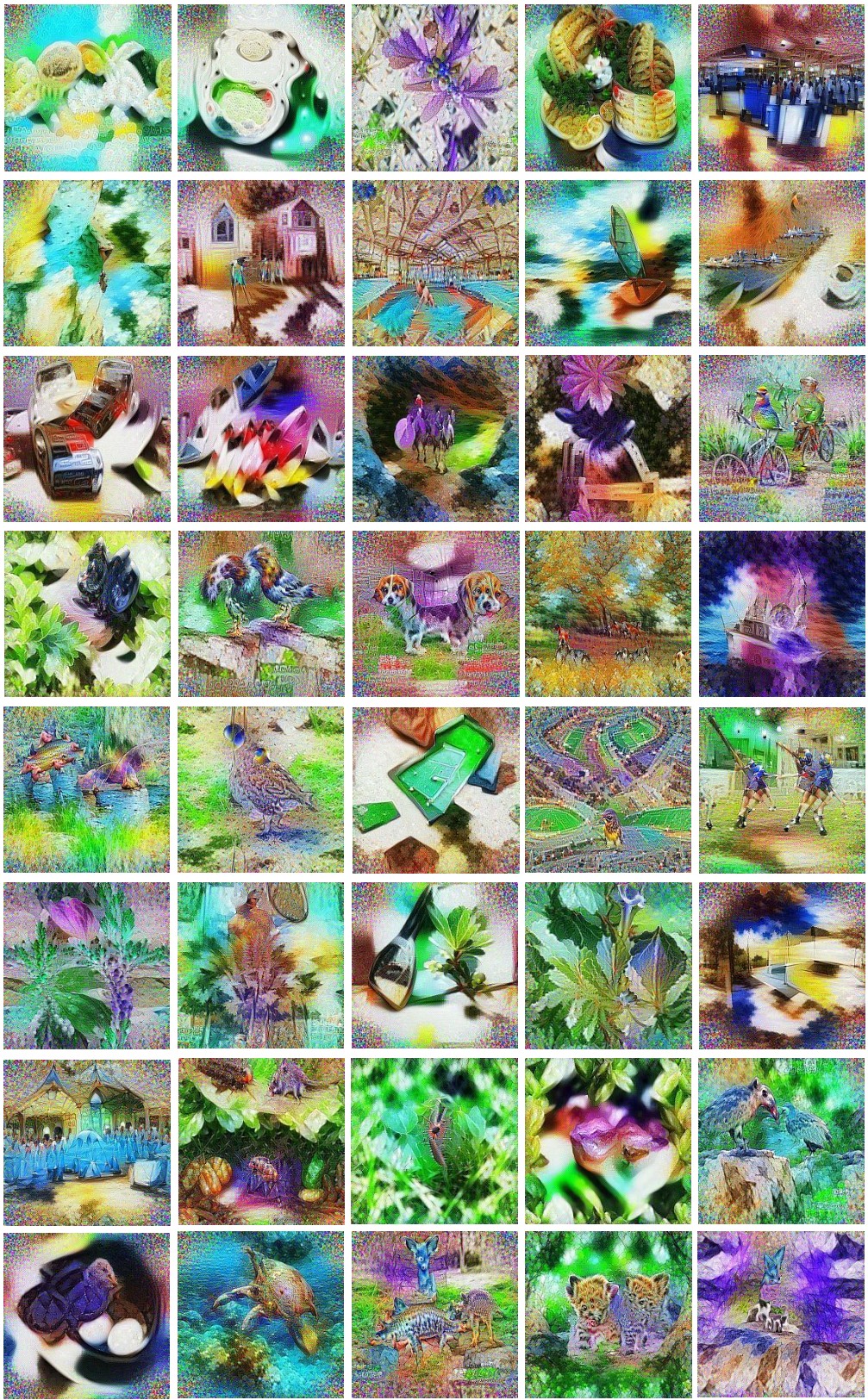

Figure 11: Synthetic ImageNet-21K data distilled from ResNet-18 by CDA.

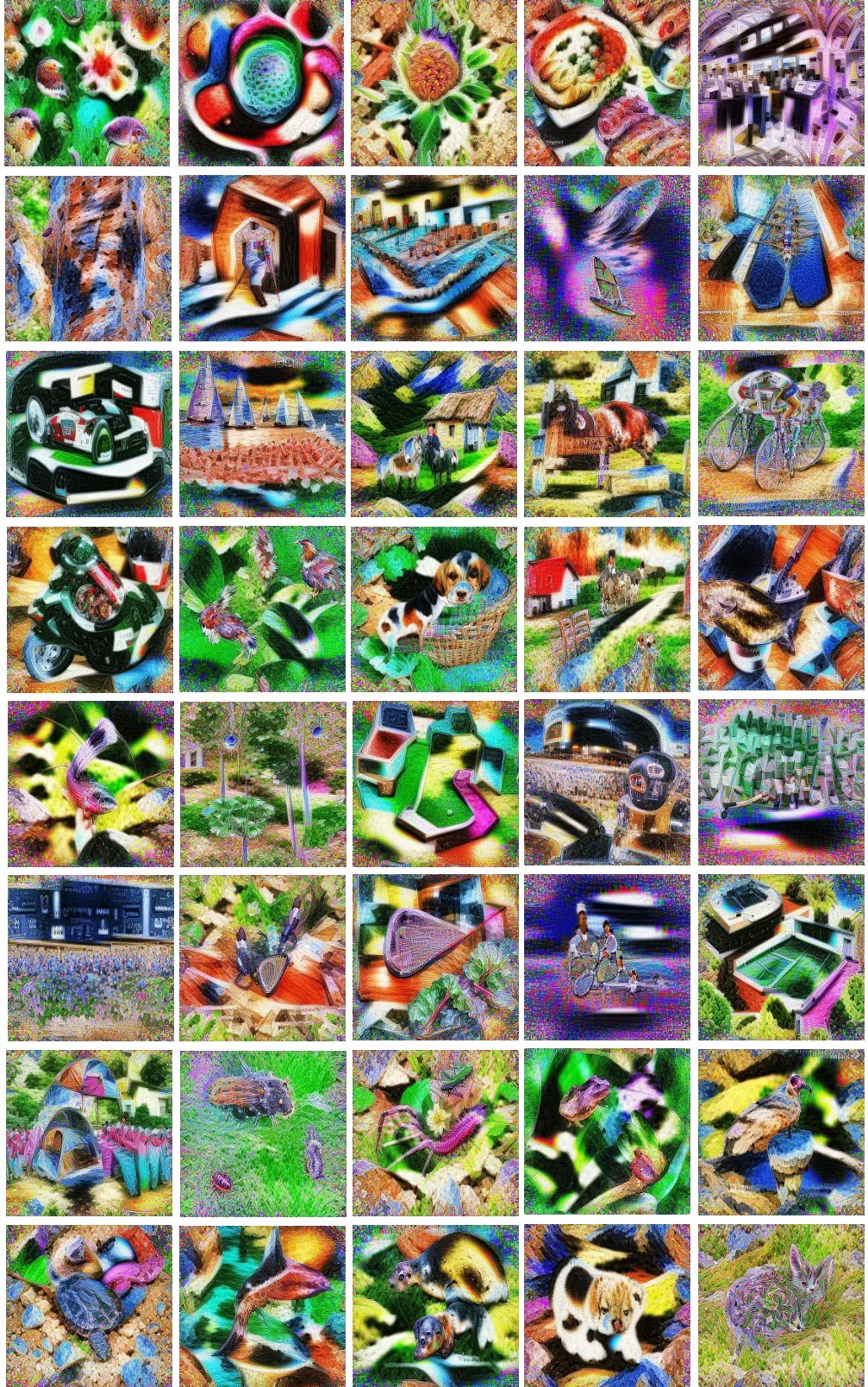

Figure 12: Synthetic ImageNet-21K data distilled from ResNet-50 by `CDA`.

