# OpenReview forum: "Dataset Distillation via Curriculum Data Synthesis in Large Data Era"
_TMLR — Accepted by TMLR_

### Review · Reviewer_2jfx · 2024-07-24

**Summary Of Contributions:**

**Claims and Contributions**: The paper introduces a method for dataset distillation with a special focus at scale using the ImageNet 21k dataset. More specifically, the authors propose curriculum data augmentation (CDA) which utilizes modulated random cropping during the synthesis stage of dataset distillation to better model the global and local structures in dataset samples. Empirically, the proposed method achieves highest accuracy on large scale ImageNet 1k and 21k datasets and outperforms competing baselines.

**Audience:**

Yes

**Claims And Evidence:**

Yes

**Requested Changes:**

1. **Rephrase**: The authors mention: “ Moreover, by working with distilled datasets, there is potential to alleviate some data privacy concerns, as raw, personally identifiable data points might be excluded from the distilled version”. This seems inaccurate as the representative subset might still violate privacy concerns. It might be worth mentioning approaches which utilize recent large scale generative models to synthesize novel samples (not a dataset subset) which have the potential to mitigate these concerns to a large extent (although these methods have their own set of challenges).

1. **Clarification on Quantitative Results**: For the quantitative results in Table 1, are the baselines also evaluated using the improved Squeeze network training as discussed in Section 3.2? If not, can the authors show an ablation of CDA applied to the ResNet 18 model for a small dataset like CIFAR-100 or Tiny ImageNet and compare it with other baselines for fair comparison?

1. The authors mention: “ We have also observed that maintaining a smaller batch size is crucial for post-training on synthetic data to achieve commendable accuracy. This is attributed to the Generalization Gap”. This is not obvious to me. Can the authors elaborate more on this in the main text?

1. While the quantitative results are good, the visualizations in Fig. 7 indicate that the quality of samples even with 2k recovery steps is not comparable to the sample quality achieved by state-of-the-art generative models on large scale datasets. I would highly recommend the authors to include a note in the conclusion highlighting some of the limitations/strengths as compared to generating datasets using powerful image generation models.

**Strengths And Weaknesses:**

**Strengths**:

1. The proposed curriculum based Random cropping is simple and works nicely at scale.
1. The quantitative evaluation is extensive which helps clarify a lot of questions.

**Weaknesses**:

**Limited methodological novelty**: Apart from the proposed data augmentation setup, most of the framework for learning distilled samples has already been proposed in prior work. Though for TMLR, I don't think this is a major problem per se.

See Requested Changes for other questions/queries.

---

> ### Author Response · Authors · 2024-08-01
> **Response to Reviewer 2jfx [1/2]**
>
> We appreciate the reviewer's constructive comments and suggestions and the recognition that the proposed method is simple, works nicely at scale, and has extensive evaluation, which helps clarify many questions. Below, we clarify each question further.
>
> > 1. Rephrase: The authors mention: “ Moreover, by working with distilled datasets, there is potential to alleviate some data privacy concerns, as raw, personally identifiable data points might be excluded from the distilled version”. This seems inaccurate as the representative subset might still violate privacy concerns. It might be worth mentioning approaches which utilize recent large scale generative models to synthesize novel samples (not a dataset subset) which have the potential to mitigate these concerns to a large extent (although these methods have their own set of challenges).
>
> Thanks for pointing this out. We appreciate the idea that large-scale generative models offer a promising way to create novel datasets without privacy concerns. However, it has yet to be explored whether these generated datasets are able to replace the original large ones. And their own challenges, like high computational costs, make it hard to apply to large dataset distillation. On the contrary, our CDA exhibits improved distillation capability and image synthesis efficiency, especially on large-scale datasets, as shown in Figure 1 and Section 4.7. We also would like to clarify that our method does not select a representative subset from the original dataset. Selecting a representative subset is more similar to a dataset pruning method. In contrast, our data distillation approach generates a new, smaller set of images. These distilled datasets are synthesized to retain the most representative information at a high-level abstraction instead of closely resembling the original dataset. Thus, it can also reduce the risk of exposing sensitive image space information from the raw data.
>
> Leveraging large-scale generative models to synthesize novel samples has been one of the directions within the dataset distillation community and has been explored in several papers [1, 2, 3]. We have discussed some of them in our related section and will expand on this discussion in our revised paper.
>
> ---
> [1] Su, Duo, Junjie Hou, Weizhi Gao, Yingjie Tian, and Bowen Tang. "D^ 4: Dataset Distillation via Disentangled Diffusion Model." In Proceedings of the IEEE/CVF Conference on Computer Vision and Pattern Recognition, pp. 5809-5818. 2024.
>
> [2] Gu, Jianyang, Saeed Vahidian, Vyacheslav Kungurtsev, Haonan Wang, Wei Jiang, Yang You, and Yiran Chen. "Efficient dataset distillation via minimax diffusion." In Proceedings of the IEEE/CVF Conference on Computer Vision and Pattern Recognition, pp. 15793-15803. 2024.
>
> [3] Cazenavette, George, Tongzhou Wang, Antonio Torralba, Alexei A. Efros, and Jun-Yan Zhu. "Generalizing dataset distillation via deep generative prior." In Proceedings of the IEEE/CVF Conference on Computer Vision and Pattern Recognition, pp. 3739-3748. 2023.
>
> ---
> > 2. Clarification on Quantitative Results: For the quantitative results in Table 1, are the baselines also evaluated using the improved Squeeze network training as discussed in Section 3.2? If not, can the authors show an ablation of CDA applied to the ResNet 18 model for a small dataset like CIFAR-100 or Tiny ImageNet and compare it with other baselines for fair comparison?
>
> Thanks for the valuable comments. The major baseline SRe$^2$L adopts the same training strategy as our CDA. Also, other baseline training is constricted to their small network due to the high computational cost and hard to scale up to ResNet-18 as a squeeze network. Therefore, for a fair comparison as much as possible, baseline results in Table 1 have been taken directly from the best evaluation performance reported by the authors in their papers. As requested, an ablation experiment result of CDA applied to the ResNet-18 model for a small dataset has been presented in Table 1. Here we provide the result comparison for CIFAR-100 and Tiny-ImageNet in the table below.
>
> |              | CIFAR-100 (IPC = 10) | CIFAR-100 (IPC = 50) | Tiny-ImageNet (IPC = 10) | Tiny-ImageNet (IPC = 50) |
> |-------|:--------:|:-------:|:--------:|:----------:|
> | DM           |                29.7 |                43.6 |                     12.9 |                     24.1 |
> | MTT          |                39.7 |                47.7 |                     23.2 |                     28.0 |
> | DataDAM      |                34.8 |                49.4 |                     18.7 |                     28.7 |
> | DATM         |                47.2 |                55.0 |                 **31.1** |                     39.7 |
> | SRe$^2$L     |                23.5 |                51.4 |                     17.7 |                     41.1 |
> | SRe$^2$L+CDA |            **49.5** |            **64.0** |                     21.3 |                 **48.7** |

---

> ### Author Response · Authors · 2024-08-01
> **Response to Reviewer 2jfx [2/2]**
>
> > 3. The authors mention: “We have also observed that maintaining a smaller batch size is crucial for post-training on synthetic data to achieve commendable accuracy. This is attributed to the Generalization Gap”. This is not obvious to me. Can the authors elaborate more on this in the main text?
>
> We elaborate on this from both empirical and theoretical perspectives:
>
> In the context of **synthetic data**, the generalization gap can be exacerbated due to the inherent differences between synthetic and real data distributions. Smaller batch sizes tend to introduce more details/noises into the gradient updates during training, which, counterintuitively, can help in better generalizing to unseen data by avoiding overfitting to the synthetic dataset's general patterns. The noise can also act as a regularizer, preventing the model from becoming too confident in its predictions on the synthetic data, which may not fully capture the complexities of large batch-size data.
>
> In Table 4, we empirically notice that utilizing a small batch size can improve model evaluation performance. This observed phenomenon aligns with the "Generalization Gap" theory, which arises when there is a lack of training samples.
>
> We will expand on this explanation in the main text to provide a clearer understanding of this observation and the underlying reasoning.
>
> > 4. While the quantitative results are good, the visualizations in Fig. 7 indicate that the quality of samples even with 2k recovery steps is not comparable to the sample quality achieved by state-of-the-art generative models on large scale datasets. I would highly recommend the authors to include a note in the conclusion highlighting some of the limitations/strengths as compared to generating datasets using powerful image generation models.
>
> Thank you for this valuable suggestion. Based on your advice, we will include a description clarifying that the quality of our generated samples is not comparable to the sample quality achieved by state-of-the-art generative models on large-scale datasets. This difference is expected, given the distinct goals of dataset distillation versus generative models. Generative models aim to synthesize highly realistic images with detailed features, whereas dataset distillation methods focus on producing images that capture the most representative information possible for efficient learning in downstream tasks. Realism is not the primary objective in dataset distillation.

---

### Review · Reviewer_G33W · 2024-08-11

**Summary Of Contributions:**

This paper addresses the challenge of dataset distillation, which involves generating a smaller yet representative subset from a large dataset to enable efficient model training while maintaining strong performance on the original testing data distribution. Previous methods have struggled with improper gradient update strategies, leading to a decline in the quality of the distilled datasets. To overcome this, the authors propose a novel global-to-local gradient refinement approach that is enhanced by curriculum data augmentation (CDA) during the data synthesis process. This method significantly improves the quality of the distilled datasets, achieving state-of-the-art accuracy on both ImageNet-1K and ImageNet-21K benchmarks.

**Audience:**

Yes

**Claims And Evidence:**

Yes

**Requested Changes:**

See the Weaknesses above.

**Strengths And Weaknesses:**

**Strengths**

* The paper is very well-written and easy to understand.

* The proposed method significantly outperforms previous dataset distillation methods.

* It is the first work to validate on a large dataset (i.e., ImageNet-21K).

* The authors perform a good analysis of their method, including an application to Continual Learning.

**Weaknesses**

The authors have done a wonderful job explaining and conveying their methods, and the method has been validated with rigorous experiments. Some weaknesses related to the formatting is as follows:

* Although it has been mentioned in related works of the different categories of previous methods, it would be great to reiterate why SRe2L and CDA are tied together in Table 1 or somewhere in the Experiments Section (Section 4).

* It would be good to highlight that Figure 4 is the central figure for explaining the proposed method, with detailed captions provided to enhance understanding.

---

> ### Author Response · Authors · 2024-08-26
> **Response to Reviewer G33W**
>
> We appreciate the reviewer's recognition and encouraging feedback. We will carefully revise our paper according to the comments provided. Below, we provide further responses to the questions raised.
>
> > Weaknesses:
> > The authors have done a wonderful job explaining and conveying their methods, and the method has been validated with rigorous experiments. Some weaknesses related to the formatting is as follows:
> > Although it has been mentioned in related works of the different categories of previous methods, it would be great to reiterate why SRe2L and CDA are tied together in Table 1 or somewhere in the Experiments Section (Section 4).
>
> Thanks for the suggestion. Our CDA follows the decoupled framework of SRe$^2$L, utilizing the *squeeze*, *recover*, and *relabel* stages for data synthesis and model updates, but incorporates curriculum design to generate data. Our experimental settings strictly match those of SRe$^2$L, ensuring a fair comparison in performance and computational cost analysis. This alignment is the reason why SRe$^2$L and CDA are closely tied together. Compared to traditional dataset distillation baselines, CDA is fundamentally different in its approach:
>
> 1. Previous methods lack scalability. The previous works like DM [1], DSA [2], and FRePo [3], work well on small-scale dataset distillation, but they are limited by the huge computational cost and cannot be scaled to large datasets and models.
> 2. Different generation paradigms. MTT [4] matches the model trajectories (weights) of training on distilled and raw datasets; RDED [5] selects and combines the raw image with diversity; D3M [6] leverages text-to-image diffusion models to generate distilled images. Thus, MTT proposed matching trajectories, RDED proposed non-optimizing, and D3M proposed diffusion-model-based generation paradigms don't belong to our knowledge-distillation-based generation approach.
> 3. Unique evaluation recipes. For instance, RDED utilizes a unique smoothed LR schedule for the learning rate reduction throughout the evaluation, which improves evaluation performance effectively.
>
> Therefore, to ensure a fair comparison as much as possible, these baseline results in Table 1 have been taken directly from the best evaluation performance reported by the authors in their papers. We will include the additional explanation and comparison details in our revised paper as suggested.
>
> [1] Bo Zhao and Hakan Bilen. Dataset condensation with distribution matching. In IEEE/CVF Winter Conference on Applications of Computer Vision, WACV 2023, Waikoloa, HI, USA, January 2-7, 2023, 2023.
>
> [2] Bo Zhao and Hakan Bilen. Dataset condensation with differentiable siamese augmentation. In International Conference on Machine Learning, pp. 12674–12685. PMLR, 2021.
>
> [3] Yongchao Zhou, Ehsan Nezhadarya, and Jimmy Ba. Dataset distillation using neural feature regression. Advances in Neural Information Processing Systems, 35:9813–9827, 2022.
>
> [4] George Cazenavette, Tongzhou Wang, Antonio Torralba, Alexei A Efros, and Jun-Yan Zhu. Dataset distillation by matching training trajectories. In Proceedings of the IEEE/CVF Conference on Computer Vision and Pattern Recognition, pp. 4750–4759, 2022a.
>
> [5] Sun, Peng, Bei Shi, Daiwei Yu, and Tao Lin. "On the diversity and realism of distilled dataset: An efficient dataset distillation paradigm." In Proceedings of the IEEE/CVF Conference on Computer Vision and Pattern Recognition, pp. 9390-9399. 2024.
>
> [6] Ali Abbasi, Ashkan Shahbazi, Hamed Pirsiavash, and Soheil Kolouri. One category one prompt: Dataset distillation using diffusion models. arXiv preprint arXiv:2403.07142, 2024.
>
>
> > It would be good to highlight that Figure 4 is the central figure for explaining the proposed method, with detailed captions provided to enhance understanding.
>
> Thank you for the valuable suggestion. We will highlight the role of Figure 4 in our revision and expand the caption to include more detailed descriptions. This will incorporate the specific *curriculum procedures in data synthesis* alongside the *Squeeze*, *Recover*, and *Relabel* stages to provide a more comprehensive overview of our dataset distillation procedure.

---

### Review · Reviewer_Rerr · 2024-08-13

**Summary Of Contributions:**

This paper proposes to improve large-scale dataset distillation using a proposed curriculum data augmentation (CDA), which gradually increases the training difficulty by lowering the minimal crop ratio.

The rest of the method is the same as a prior method SRe$^2$L. There are 3 steps:
- **Squeeze**, where a model is trained on the real dataset and frozen in the following stages. This step follows from the practices in SRe$^2$L.
- **Recover**, which is the synthesizing step where images are generated using gradient information. The motivation for CDA is that we want to refine the gradients from global to local throughout training.
  - For CDA, the paper considers 2 scheduling for the curriculum (i.e. linear and cosine), and searches over the length of the curriculum (called the "milestone").
- **Relabel**, where a model is . The paper finds that small-batch training helps improve the performance, especially when the data is limited.

The paper shows that CDA improves over the baseline SRe$^2$L across datasets (CIFAR-100, Tiny-ImageNet, ImageNet-1K and 21K) and models (ResNet-18, 50, 101, DenseNet, RegNet, ConvNeXt, DeiT). The paper provides ablation study on the effects of curriculum scheduling and batch size.

**Audience:**

Yes

**Broader Impact Concerns:**

There are no direct societal or ethical concerns.

**Claims And Evidence:**

Yes

**Requested Changes:**

Questions:
- In "Synthesis Cost", could you add a comment comparing the synthesis time to the training time of the models being tested?
- We may want to trade off quality versus efficiency. Could you comment on how the quality of the pretrained model and the number of recovery iterations affect the quality of the generated data? For example, why Table 7(b) uses 1000 recover iterations?
- Could you comment on potential ways to reduce the synthesis cost? For example, would selecting (rather than randomly sampling) the images per class provide improvements?
- Table 1: What would the results be for higher IPC (e.g. 32%, 64%)? I’m wondering whether there’s a smooth growth in performance w.r.t. IPC. Please treat this question as optional since it's expensive to generate more images.

**Strengths And Weaknesses:**

Strength:
- Although CDA is the only major technical modification, it is effective and works well across datasets and architectures. It's interesting to know that a simple change in curriculum suffices to provide good performance.
- The paper provides thorough experiments and ablation studies.
  - This includes comparison at various Image-per-class (IPC), curriculum scheduling (or ranges for constant cropping ratios), and batch sizes.
- It discusses the computation cost for synthesis.

Weakness: My main complaint is that the method is highly specific to the baseline SRe$^2$L, which is the only baseline in this work.
- The paper mentions several related works at the end of the related work section. Could you provide a brief description on these works, and the reason why these methods are not compared against?
- Are there other baselines that can achieve comparable performance, and if yes, can the curriculum idea be applied to these baselines?

Some other comments:
- The title should be made more specific. Currently the title is overly broad and vague.
- Eq (3): why taking the sup rather than the average?
- In the paragraph of "Relabel: Post-training on Larger Models with Stronger Training Recipes.", the paper mentions that another prior method TESLA has a decline in accuracy when scaled up. Could you comment on why?
- Fig 4: Each step has 2 boxes, one with solid borders, one with dashed borders. Are the two boxes of the same type (i.e. simply 2 crops), or does the border type signal anything?
- At the end of section 3, the paragraph of "Advantages of Global-to-local Synthesis": this paragraph mentions 3 advantages of the proposed CDA: (1) stabilized training, (2) better generalization, and (3) avoiding overfitting. Aren't "better generalization" and "avoiding overfitting" the same?
- Are $\beta_l, \beta_u$ used other than in Table 2? If not, please remove these notations and use 0.08, 1 directly.
- The ablation on the curriculum schedule (i.e. Fig 6): I wonder how much difference linear vs cosine schedules have -- how many seeds are used in producing the bars in Fig 6? Also, it seems that the best performance is from Linear at 90%.
- There's a minor typo next to Table 6.

---

> ### Author Response · Authors · 2024-08-26
> **Response to Reviewer Rerr [1/4]**
>
> We appreciate the detailed reviews, constructive comments and suggestions from the reviewer. We will accommodate all the feedback in our revised version. In the following, we clarify each question in detail for the reviewer.
>
> > Weakness: My main complaint is that the method is highly specific to the baseline SRe$^2$L, which is the only baseline in this work.
>
> Thanks for raising this concern. Our method is designed to be applicable to general decoupled dataset distillation approaches. To demonstrate this, we apply our approach to the self-supervised SC-DD [1], achieving 1.0 and 0.9 improvements on Tiny-ImageNet and ImageNet-1K, respectively, as shown in the table below. In addition to SRe$^2$L, we have also included several well-known dataset distillation methods as baselines in Table 1 of our original submission, such as DM [2], DSA [3], FRePo [4], MTT [5], DataDAM [6], TESLA [7], and DATM [8].
>
> | Method      | Tiny-ImageNet | ImageNet-1K |
> |---|:---:|:----:|
> | SC-DD [1]    |        45.5  |       53.1 |
> |SC-DD + Curriculum (Ours) |     &nbsp;&ensp; 46.5$^{\uparrow1.0}$ |    &nbsp;&ensp; 54.0$^{\uparrow0.9}$ |
>
> [1] Zhou, M., Yin, Z., Shao, S., & Shen, Z. (2024). Self-supervised Dataset Distillation: A Good Compression Is All You Need. arXiv preprint arXiv:2404.07976.
>
> [2] Bo Zhao and Hakan Bilen. Dataset condensation with distribution matching. In IEEE/CVF Winter Conference on Applications of Computer Vision, WACV 2023, Waikoloa, HI, USA, January 2-7, 2023, 2023.
>
> [3] Bo Zhao and Hakan Bilen. Dataset condensation with differentiable siamese augmentation. In International Conference on Machine Learning, pp. 12674–12685. PMLR, 2021.
>
> [4] Yongchao Zhou, Ehsan Nezhadarya, and Jimmy Ba. Dataset distillation using neural feature regression. Advances in Neural Information Processing Systems, 35:9813–9827, 2022.
>
> [5] George Cazenavette, Tongzhou Wang, Antonio Torralba, Alexei A Efros, and Jun-Yan Zhu. Dataset distillation by matching training trajectories. In Proceedings of the IEEE/CVF Conference on Computer Vision and Pattern Recognition, pp. 4750–4759, 2022a.
>
> [6] Sajedi, Ahmad, Samir Khaki, Ehsan Amjadian, Lucy Z. Liu, Yuri A. Lawryshyn, and Konstantinos N. Plataniotis. "Datadam: Efficient dataset distillation with attention matching." In Proceedings of the IEEE/CVF International Conference on Computer Vision, pp. 17097-17107. 2023.
>
> [7] Cui, Justin, Ruochen Wang, Si Si, and Cho-Jui Hsieh. "Scaling up dataset distillation to imagenet-1k with constant memory." In International Conference on Machine Learning, pp. 6565-6590. PMLR, 2023.
>
> [8] Guo, Ziyao, Kai Wang, George Cazenavette, HUI LI, Kaipeng Zhang, and Yang You. "Towards Lossless Dataset Distillation via Difficulty-Aligned Trajectory Matching." In The Twelfth International Conference on Learning Representations, 2024.
>
> > W1: The paper mentions several related works at the end of the related work section. Could you provide a brief description on these works, and the reason why these methods are not compared against?
>
> Thanks for the suggestion. Our consideration and motivation of the related work section is that we aim to reference survey-like works, including a comprehensive study and literature list [1, 2, 3] on framework design space [4] and adversarial robustness benchmarks [5] in dataset distillation. For the comparison, these methods each employ distinct dataset distillation paradigms and evaluation strategies, or have different distillation goals, making direct comparison unsuitable. Specifically, RDED [1] selects and combines raw image patches to enhance diversity, and D3M [2] uses text-to-image diffusion models for generating distilled images. These approaches are substantially different from our input-optimization-based approach. Additionally, GUARD [3] incorporates curvature regularization to embed adversarial robustness, focusing on a different objective than our CDA. As suggested, we will include brief descriptions on these methods and the criteria for selecting baselines in our revised paper.
>
> [1] Peng Sun, Bei Shi, Daiwei Yu, and Tao Lin. "On the diversity and realism of distilled dataset: An efficient dataset distillation paradigm." In Proceedings of the IEEE/CVF Conference on Computer Vision and Pattern Recognition, pp. 9390-9399. 2024.
>
> [2] Ali Abbasi, Ashkan Shahbazi, Hamed Pirsiavash, and Soheil Kolouri. One category one prompt: Dataset distillation using diffusion models. arXiv preprint arXiv:2403.07142, 2024.
>
> [3] Eric Xue, Yijiang Li, Haoyang Liu, Yifan Shen, and Haohan Wang. Towards adversarially robust dataset distillation by curvature regularization. arXiv preprint arXiv:2403.10045, 2024.
>
> [4] Shitong Shao, Zikai Zhou, Huanran Chen, and Zhiqiang Shen. Elucidating the design space of dataset condensation. arXiv preprint arXiv:2404.13733, 2024b.
>
> [5] Yifan Wu, Jiawei Du, Ping Liu, Yuewei Lin, Wenqing Cheng, and Wei Xu. Dd-robustbench: An adversarial robustness benchmark for dataset distillation. arXiv preprint arXiv:2403.13322, 2024.

---

> ### Author Response · Authors · 2024-08-26
> **Response to Reviewer Rerr [2/4]**
>
> > W2: Are there other baselines that can achieve comparable performance, and if yes, can the curriculum idea be applied to these baselines?
>
> We present the results of these approaches with an IPC of 50 in the table below for your reference. For the application of CDA to the baseline, we apply CDA to SC-DD baseline. As shown in the table below, the results indicate that our CDA improves SC-DD by  0.9% on ImageNet-1K.
>
> | Method     | CIFAR-100 | ImageNet-1K |
> |------------|:----------:|:------------:|
> | CDA (ours) |      64.0 |          53.5 |
> | D3M [1]       |      54.5 |       32.2 |
> | GUARD [2]     |        -- |            39.9 |
> | SC-DD [3]     |    --   |         53.1 |
> |SC-DD [3] + Curriculum (Ours) |       --  |        &nbsp;&nbsp; 54.0$^{\uparrow0.9}$ |
>
> [1] Ali Abbasi, Ashkan Shahbazi, Hamed Pirsiavash, and Soheil Kolouri. One category one prompt: Dataset distillation using diffusion models. arXiv preprint arXiv:2403.07142, 2024.
>
> [2] Eric Xue, Yijiang Li, Haoyang Liu, Yifan Shen, and Haohan Wang. Towards adversarially robust dataset distillation by curvature regularization. arXiv preprint arXiv:2403.10045, 2024.
>
> [3] Zhou, Muxin, Zeyuan Yin, Shitong Shao and Zhiqiang Shen. “Self-supervised Dataset Distillation: A Good Compression Is All You Need.” ArXiv abs/2404.07976 (2024).
>
> > Some other comments:
> > C1: The title should be made more specific. Currently the title is overly broad and vague.
>
> Thanks for the suggestion. We will update the title to "Dataset Distillation via Curriculum Data Synthesis in Large Data Era". Our original title was chosen for several reasons. Firstly, our proposed CDA represents a pioneering approach that successfully distills the ImageNet-21K training dataset with a 50$\times$ compression and less than a 15% absolute accuracy loss. Secondly, this significant performance demonstrates its effectiveness and efficiency in distilling large-scale datasets, marking it as a milestone in dataset distillation for the large data era.
>
> > C2: Eq (3): why taking the sup rather than the average?
>
> In the field of dataset distillation, we may be more concerned with the model's performance on the data samples in the worst-case scenario rather than the average performance since the worst-case scenario serves as a key indicator of the model's ability to differentiate and handle difficult cases in the validation set when trained on the distilled dataset.
>
> > C3: In the paragraph of "Relabel: Post-training on Larger Models with Stronger Training Recipes.", the paper mentions that another prior method TESLA has a decline in accuracy when scaled up. Could you comment on why?
>
> The trajectory-based matching approaches, MTT and TESLA, generate images by excessively optimizing to align the dense training trajectories of model weights at each epoch between real and distilled datasets on a specific backbone model. As a result, the distilled dataset becomes overly dependent on this model, potentially leading to overfitting and reduced effectiveness when training other models, particularly larger ones.
>
> > C4: Fig 4: Each step has 2 boxes, one with solid borders, one with dashed borders. Are the two boxes of the same type (i.e. simply 2 crops), or does the border type signal anything?
>
> Thanks for your careful review. We crop the image with only one box region each step, and we plot another box with dashed borders to indicate a random or flexible selection procedure of the box location. We will make this clearer in the caption of Figure 4 in our revision.
>
> > C5: At the end of section 3, the paragraph of "Advantages of Global-to-local Synthesis": this paragraph mentions 3 advantages of the proposed CDA: (1) stabilized training, (2) better generalization, and (3) avoiding overfitting. Aren't "better generalization" and "avoiding overfitting" the same?
>
> They are not the same. "Better generalization" refers to the ability of models trained on the distilled datasets to perform well across a wider range of evaluation scenarios. However, "avoiding overfitting" specifically refers to our curriculum strategy during the distillation process, where we use a flexible region update in each iteration to prevent overfitting that could occur with a fixed region update. We will clarify the distinction between these two advantages in the revision.
>
> > C6: Are $\beta_l$, $\beta_u$ used other than in Table 2? If not, please remove these notations and use 0.08, 1 directly.
>
> Thanks for the suggestion. $\beta_l$ and $\beta_u$ mainly appear in three places: Algorithm 1, Table 2, and Appendix B.1. They represent the default lower and upper bounds of crop scale, set to 0.08 and 1 unless otherwise specified. Notably, in Appendix B.1, the value of $\beta_l$ is changed for the adaptation to CIFAR-100. We will revise this part to make them clearer.

---

> ### Author Response · Authors · 2024-08-27
> **Response to Reviewer Rerr [3/4]**
>
> > C7: The ablation on the curriculum schedule (i.e. Fig 6): I wonder how much difference linear vs cosine schedules have -- how many seeds are used in producing the bars in Fig 6? Also, it seems that the best performance is from Linear at 90%.
>
> We present the detailed value comparison between linear and cosine schedulers in Fig.6. Each number is obtained by averaging repeated experiments with three different seeds. The table below shows that the difference between linear and cosine schedulers is marginally slight and the best result for linear is at the milestone of 90% while the cosine scheduler performs better in the end. To avoid manually setting the milestone percentage for the linear scheduler, we adopt the cosine scheduler with a milestone percentage of 100% in our experiments.
>
> | Milestone (\%) |    10  |    20  |    30  |    40 |    50 |    60 |    70 |    80 |    90 |   100 |
> |--------|:------:|:------:|:------:|:------:|:------:|:------:|:------:|:------:|:------:|:------:|
> | Linear    | 45.39 | 46.30 | 46.59 | 46.51 | 46.60 | 47.18 | 47.13 | 47.37 | 48.05 | 47.78 |
> | Consine   | 45.41 | 45.42 | 46.15 | 46.89 | 46.93 | 47.42 | 46.86 | 47.33 | 47.80 | 48.05 |
>
> > C8: There's a minor typo next to Table 6.
>
> Thanks for pointing it out. The correct statement is "we evaluate the Top-1 accuracy on CDA, SRe$^2$L and real ImageNet-1K training set, using a mean of random 10-crop and global images. " We will fix this typo in our revision.
>
> > Questions: Q1: In "Synthesis Cost", could you add a comment comparing the synthesis time to the training time of the models being tested?
>
> Thanks for the suggestion. Typically, the synthesis process can be a little bit more computationally intensive and time-consuming compared to the training of models on the distilled data. For instance, while the training of a model on a distilled dataset might take a few hours, the synthesis process could take longer with more hours, depending on the IPCs of the data and the number of iterations required. Specifically, for ImageNet-1K, it takes about 29 hours to generate the distilled ImageNet-1K with 50 IPC on a single A100 (40G) GPU and the peak GPU memory utilization is 6.7GB. For ImageNet-21K, it takes 11 hours to generate ImageNet-21K images 1 IPC on a single RTX 4090 GPU and the peak GPU memory utilization is 15GB. In our experiment, it takes about 55 hours to generate the entire distilled ImageNet-21K with 20 IPC on 4$\times$ RTX 4090 GPUs in total. While, once the dataset is synthesized, the same dataset can be reused for training multiple models, potentially amortizing the synthesis cost over multiple experiments. In our work, the synthesis time refers to the time taken to generate the distilled dataset, the training time of the tested model depends on the model's architecture, parameters, and FLOPs. We present the detailed training time of several tested models, as shown in the table below.
>
> | Model            | ResNet-18 | ResNet-50  | ResNet-101 | DenseNet-121 | RegNet-Y-8GF |  ConvNeXt-Tiny  | DeiT-Tiny |
> |---------------|:----------:|:-----------:|:-----------:|:----------:|:-------:|:--------:|:------:|
> | training time (A100 GPU hours) |       2.3 |        7.1 |       12.5 |          9.1 |         13.6 |            18.3 |       4.6 |
>
> > Q2: We may want to trade off quality versus efficiency. Could you comment on how the quality of the pretrained model and the number of recovery iterations affect the quality of the generated data? For example, why Table 7(b) uses 1000 recover iterations?
>
> Higher-quality pre-trained models and longer recovery iterations generally lead to better quality generated data. As shown in the additional `SC-DD + Curriculum (Ours)` experiments in *Weakness*, the compressed dataset distilled from a self-supervised pre-trained model with higher quality achieves higher validation performance. Regarding recovery iterations, we follow the SRe$^2$L settings to use 1000 iterations to recover CIFAR-100.

---

> ### Author Response · Authors · 2024-08-27
> **Response to Reviewer Rerr [4/4]**
>
> > Q3: Could you comment on potential ways to reduce the synthesis cost? For example, would selecting (rather than randomly sampling) the images per class provide improvements?
>
> We appreciate your suggestion to reduce synthesis costs, which is crucial to consider. Selecting representative real images for input data initialization instead of the Gaussian noise initialization will be an effective way to accelerate image distillation and reduce synthesis costs. It can potentially reduce the number of recovery iterations required to achieve high-quality distilled data and this approach could lead to faster convergence and lower synthesis costs.
>
> > Q4: Table 1: What would the results be for higher IPC (e.g. 32%, 64%)? I’m wondering whether there’s a smooth growth in performance w.r.t. IPC. Please treat this question as optional since it's expensive to generate more images.
>
> Given the increasing performance under the IPC settings among 10, 50, and 200, models learned from distilled datasets with higher IPC are expected to have improved performance. As the IPC increases, the model has access to more information, allowing it to better capture the data distribution and thus improve its performance. However, this improvement is typically subject to diminishing returns, after a certain point, additional increases in IPC may result in smaller performance gains relative to the increased computational cost. This behavior is typical in scenarios where more data leads to better generalization, but each additional data point contributes less incremental value. Another drawback of higher IPC is the cost increase in the synthesis and post-training phases due to the involvement of more data.

---

### Author Response · Authors · 2024-10-03
**A revision of our manuscript has been updated.**

We extend our gratitude to all the reviewers for your invaluable comments. We have diligently prepared a thorough response to address all your concerns and updated the manuscript accordingly to your suggestions.

We are encouraged by the positive comments from reviewers, including the paper is very well-written and easy to understand / the proposed curriculum based Random cropping is simple and works nicely at scale [Reviewer G33W, 2jfx]; CDA significantly outperforms previous dataset distillation methods / is effective and works well across datasets and architectures / quantitative evaluation is extensive which helps clarify a lot of questions [Reviewer Rerr, G33W, 2jfx]; The first work to validate on a large dataset (i.e., ImageNet-21K) [Reviewer G33W]; The paper provides thorough experiments and ablation studies / perform a good analysis of their method, including an application to Continual Learning [Reviewer Rerr, G33W, 2jfx]; It discusses the computation cost for synthesis [Reviewer Rerr].

We also thank the reviewers for constructive comments, such as providing a brief description on the related works [Reviewer Rerr], reiterate why SRe2L and CDA are tied together in Table 1 or somewhere in the Experiments Section (Section 4) [Reviewer G33W], rephrase and clarification / include a note in the conclusion highlighting some of the limitations/strengths [Reviewer 2jfx]. We have accommodated all of the comments in our revised manuscripts, where all the revised contents are highlighted in $\textcolor{blue}{\text{blue}}$ color.

Detailed responses to each reviewer's questions, concerns, and requested changes are provided in the following replies. We are eager for you to explore our detailed responses. We always welcome further discussion, and your insights are crucial to refining our paper.

---

### Decision · Action_Editor_7nEz · 2024-10-13

**Recommendation:** Accept as is

**Comment:**

After the authors' revision, most of the reviewers' concerns have been addressed. The reviewers unanimously recommend acceptance of the paper.

I find the work valuable and recommend its acceptance and publication.

**Audience:**

Researchers and practitioners focused on dataset distillation or condensation, as well as those dealing with model training under constraints like limited computational and storage resources or data privacy concerns, may find this paper of interest.

**Claims And Evidence:**

Summary:

This paper proposes curriculum data augmentation (CDA), a novel, scalable dataset distillation approach that synthesizes a small, representative subset from a large dataset while preserving competitive performance in models trained on the subset. The key idea leverages modulated random cropping during the dataset distillation process to better capture both global and local structures within dataset samples in a progressive manner. Evaluations on widely-used benchmarks and the large-scale ImageNet datasets demonstrate the effectiveness of the proposed CDA.


Claims:

The key claims made in the paper are that (1) the proposed global-to-local gradient update strategy is effective for dataset distillation; (2) the resulting CDA method outperforms previous approaches on various datasets, and in particular facilitates large-scale dataset distillation; and (3) the synthetic data are useful for downstream tasks.

Evidence:

The claims are well supported by the characteristics of the proposed method, along with the experimental results presented in both the original submission and the revision.